# MRAD: Zero-Shot Anomaly Detection with Memory-Driven Retrieval

**Chaoran Xu[1,2], Chengkan Lv[1,2], Qiyu Chen[1,2], Feng Zhang[1,2],\* Zhengtao Zhang[1,2]**
[1]Institute of Automation, Chinese Academy of Sciences, Beijing, China
[2]School of Artificial Intelligence, University of Chinese Academy of Sciences, Beijing, China
`{xuchaoran2024, chengkan.lv, chenqiyu2021,`
`zhengtao.zhang, feng.zhang}@ia.ac.cn`

## Abstract

Zero-shot anomaly detection (ZSAD) often leverages pretrained vision or vision-language models, but many existing methods use prompt learning or complex modeling to fit the data distribution, resulting in high training or inference cost and limited cross-domain stability. To address these limitations, we propose Memory-Retrieval Anomaly Detection method (MRAD), a unified framework that replaces parametric fitting with a direct memory retrieval. The train-free base model, MRAD-TF, freezes the CLIP image encoder and constructs a two-level memory bank (image-level and pixel-level) from auxiliary data, where feature-label pairs are explicitly stored as keys and values. During inference, anomaly scores are obtained directly by similarity retrieval over the memory bank. Based on the MRAD-TF, we further propose two lightweight variants as enhancements: (i) MRAD-FT fine-tunes the retrieval metric with two linear layers to enhance the discriminability between normal and anomaly; (ii) MRAD-CLIP injects the normal and anomalous region priors from the MRAD-FT as dynamic biases into CLIP's learnable text prompts, strengthening generalization to unseen categories. Across 16 industrial and medical datasets, the MRAD framework consistently demonstrates superior performance in anomaly classification and segmentation, under both train-free and training-based settings. Our work shows that fully leveraging the empirical distribution of raw data, rather than relying only on model fitting, can achieve stronger anomaly detection performance. The code will be publicly released at `https://github.com/CROVO1026/MRAD`.

## 1 Introduction

Anomaly detection (AD) seeks to identify regions or samples that deviate substantially from normal patterns (Pang et al., 2021; Ruff et al., 2021), with broad impact in industrial inspection (Bergmann et al., 2019; 2020; Chen et al., 2024a; 2025a; Qu et al., 2023) and medical imaging (Qin et al., 2022; Fang et al., 2025b). Labeled data in the target domain is often scarce or unavailable, motivating interest in ZSAD (Zhou et al., 2024; Chen et al., 2025b). The key challenge is to bridge the semantic gap between generic pretrained features and domain-specific anomaly patterns in the absence of target-domain supervision, while preserving both efficiency and accuracy (Fang et al., 2025a).

Vision-language pretrained models (VLMs), such as CLIP (Radford et al., 2021), learn powerful cross-modal representations via contrastive training on large-scale image-text pairs, enabling new possibilities for ZSAD. CLIP-based ZSAD approaches fall into two paradigms. *Prompt-ensemble* (Guo et al., 2023b; Li et al., 2022) methods employ hand-crafted templates (e.g., "normal object," "defective object") to match visual features without training, as in WinCLIP (Jeong et al., 2023) and CLIP-AD (Chen et al., 2024b). However, their generalization is constrained by prompt set size/diversity and domain expertise, and they often struggle in complex scenes. *Prompt-optimization* (Guo et al., 2023a) methods replace or augment text prompts with learnable vectors for domain adaptation. For instance, AnomalyCLIP (Zhou et al., 2024) introduces object-agnostic prompts and a diagonal prominent attention map (DPAM) to strengthen local anomaly detection;

---

*Corresponding author.

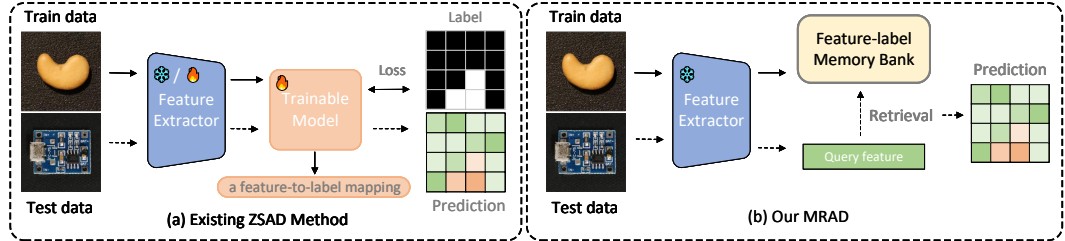

Figure 1: Most existing ZSAD methods parameterize and fit $p(y \mid x)$ solely via a trainable model, which may cause information loss. Our MRAD is to directly access the empirical training distribution by retrieving from a feature-label memory bank. The dashed line denotes the inference process.

AdaCLIP (Cao et al., 2024) and FAPrompt (Zhu et al., 2025) combine static and dynamic prompts to enhance cross-domain generalization.

While recent works make great progress, several important limitations persist: (1) Prompt learning, feature engineering, and vision-branch fine-tuning increase architectural complexity, which weakens generalization and raises computation cost. (2) Relying solely on auxiliary learners (e.g., prompt learners or MLPs) to fit the conditional distribution $p(y \mid x)$ may result in information loss. (3) dynamic prompt composition is sensitive to the source of "dynamic" information, which materially affects pixel-level segmentation and vision-text alignment.

Our recent analysis across datasets shows that similarity to sets of normal and abnormal features reliably reflects the degree of normality while preserving subtle intrinsic variations (see section 3.1.1). Motivated by this observation, we propose a novel ZSAD framework MRAD, which replaces parametric fitting with a direct feature-label retrieval paradigm as shown in Figure 1. Specifically, we construct a two-level memory bank that explicitly stores feature-label pairs as keys and values. During inference, anomaly scores are obtained via similarity retrieval, thereby reducing training cost, overfitting, and information loss. Based on this train-free MRAD model (MRAD-TF), we propose two lightweight enhancements: (i) MRAD-FT, which only adds two linear layers to fine-tune the retrieval metric, improves both classification and segmentation at low training cost; (ii) MRAD-CLIP, which injects region priors from MRAD-FT into learnable CLIP text prompts as dynamic biases, enhances localization and generalization to unseen categories. This variant improves upon conventional dynamic prompt learning, which shows insufficient effectiveness in unseen categories.

In summary, this paper makes the following main contributions:

- We present MRAD, a simple yet effective framework for ZSAD. Built upon a two-level memory bank, MRAD-TF achieves competitive detection performance under a training-free setting via similarity retrieval. Furthermore, with a few trainable parameters, the fine-tuned MRAD-FT variant can significantly enhance the discriminative ability.

- We further propose the MRAD-CLIP variant, which optimizes traditional CLIP-based prompt learning methods by injecting normal and anomalous region priors from MRAD-FT into learnable prompts. This guides the model to focus on anomalous regions, improves its performance on unseen categories, and enables precise local anomaly detection.

- Trained on a single auxiliary dataset, MRAD variants outperform existing state-of-the-art methods (e.g., AnomalyCLIP, FAPrompt) on 16 industrial and medical datasets, demonstrating superior cross-domain generalization and robustness.

## 2 RELATED WORK

**CLIP and Fine-tuning Adaptation.** CLIP (Radford et al., 2021) aligns visual and textual representations via contrastive learning on large-scale image-text pairs, delivering strong cross-modal representations and zero-shot transfer, yet performance on downstream tasks still hinges on prompt design. To improve adaptation, CoOp (Zhou et al., 2022b) replaces prompt context words with learnable vectors to automatically optimize prompts and CoCoOp (Zhou et al., 2022a) adds an image-

conditioned encoder to generate dynamic prompts for better generalization. Tip-Adapter (Zhang et al., 2022) caches and fuses features to adapt quickly without updating the backbone, improving few-shot performance. Building on CLIP, many ZSAD methods adopt CoOp-style prompt learning, which partly reduces reliance on manual prompt engineering. However, challenges persist in cross-domain generalization and in precisely modeling normal/abnormal semantics, especially at the pixel level. Our core idea is closer in spirit to Tip-Adapter, caching raw features and making retrieval-based decisions, but ZSAD is open-set rather than closed-set classification, which makes it challenging to build and maintain a cache that covers both normal and anomalous distributions.

**Zero-shot Anomaly Detection.** ZSAD can leverage large vision-language models to enable cross-category, supervision-free detection, typically aligning image features with normal/abnormal semantics using manual or learnable prompts (Gu et al., 2024; Qu et al., 2024). Early works such as WinCLIP (Jeong et al., 2023), APRIL-GAN (Chen et al., 2023), and CLIP-AD (Chen et al., 2024b) rely on hand-crafted prompt templates, limiting generalization. To enhance anomaly awareness, methods like AnomalyCLIP (Zhou et al., 2024) and AdaCLIP (Cao et al., 2024) introduce learnable context tokens to mitigate cross-domain semantic bias and improve domain-specific adaptation; FAPrompt (Zhu et al., 2025) further refines dynamic prompt design while retaining a similar training/inference paradigm. More recent methods, such as AA-CLIP (Ma et al., 2025) with anomaly-aware textual anchors and Bayes-PFL (Qu et al., 2025) with bayesian prompt modeling plus residual cross-model attention, pursue broader prompt-space coverage and finer feature alignment. Despite these advances, most methods still only fit the data-label distribution with a single newly trained component, which will lead to information loss and reduced generalization.

## 3 MRAD: MEMORY-RETRIEVAL ANOMALY DETECTION

### 3.1 OVERVIEW

#### 3.1.1 EXPLANATION OF OUR MOTIVATION

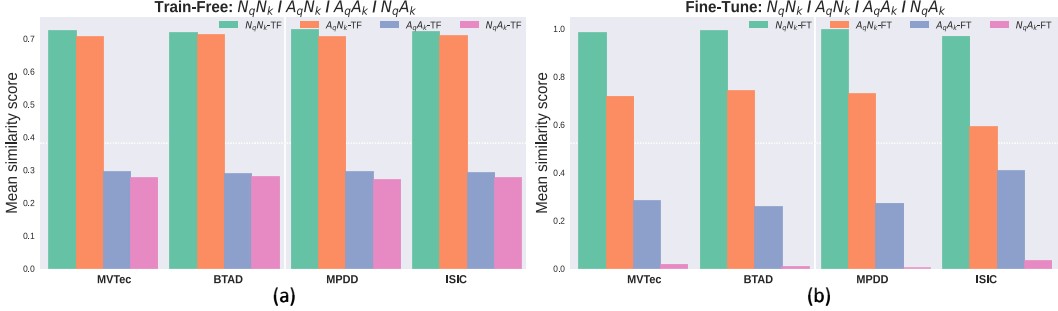

Figure 2: Mean similarity scores for different query-key relations (queries from four datasets; keys and values from VisA). **(a)** Train-free setting: the consistent ordering $N_q N_k > A_q N_k$ and $A_q A_k > N_q A_k$ shows that similarity to normal/abnormal keys provides stable discriminative signals. **(b)** Fine-tune setting: a lightweight fine-tuning further enlarges the margin between $A_q A_k$ and $N_q A_k$, demonstrating that fine-tuning improves separability while preserving the same ordering.

Based on the similarity distributions of a frozen CLIP image encoder, we conduct a cross-dataset study by taking patch features from dataset-a as queries and patch features from dataset-b as keys. Let $K$ be the set of anomalous and normal keys; let $V$ be a one-hot label matrix aligned with the keys. For a query feature $q$, we define its retrieval-based anomaly and normal similarity scores as

$$s_{\mathrm{anom}}(q) = \left[ \mathrm{softmax}\big(qK^{\top}/\tau\big) V \right]_a, \tag{1}$$

$$s_{\mathrm{norm}}(q) = \left[ \mathrm{softmax}\big(qK^{\top}/\tau\big) V \right]_n = 1 - s_{\mathrm{anom}}(q). \tag{2}$$

where $\tau$ is the temperature, $[\cdot]_a$ selects the "anomalous" channel. In words, $s_{\mathrm{anom}}(q)$ is the softmax similarity score that $q$ assigns to anomalous keys, while $s_{\mathrm{norm}}(q)$ is assigned to normal keys. Then

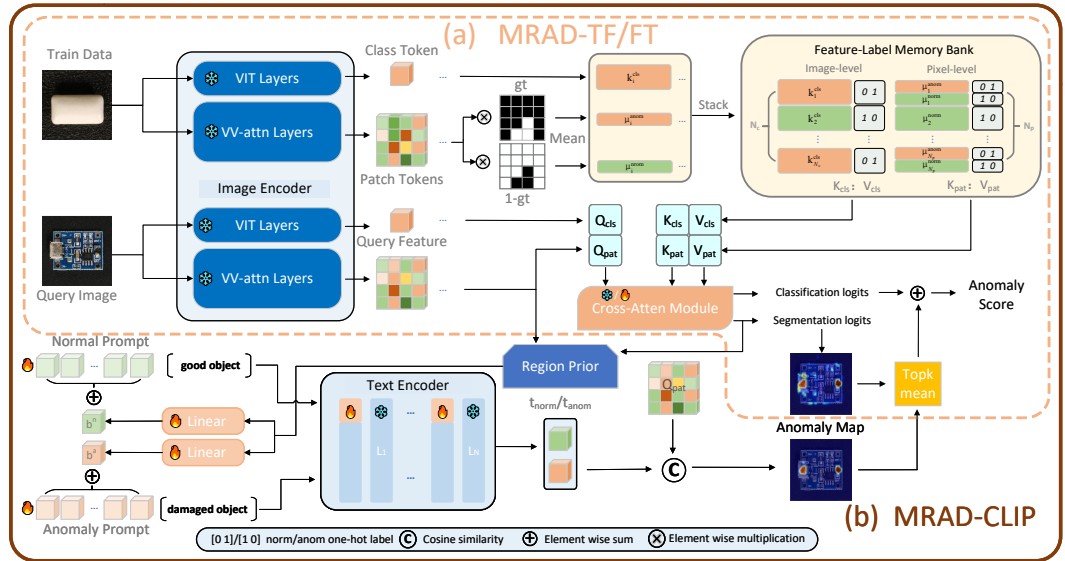

Figure 3: Overall architecture of the proposed MRAD framework. **(a) MRAD-TF/FT**: query features from a frozen CLIP image encoder are matched to a two-level feature-label memory bank; MRAD-TF uses direct similarity retrieval, while MRAD-FT adds linear layers into cross-atten module. **(b) MRAD-CLIP**: priors from MRAD-FT are injected as dynamic biases into learnable text prompts, enhancing cross-modal alignment and anomaly localization.

let $\Omega_A$ and $\Omega_N$ denote the index sets of anomalous and normal queries, we aggregate four dataset-level statistics as follows:

$$A_q A_k = \frac{1}{|\Omega_A|} \sum_{u \in \Omega_A} s_{\mathrm{anom}}(q_u), \qquad N_q A_k = \frac{1}{|\Omega_N|} \sum_{u \in \Omega_N} s_{\mathrm{anom}}(q_u), \qquad (3)$$

$$A_q N_k = \frac{1}{|\Omega_A|} \sum_{u \in \Omega_A} s_{\mathrm{norm}}(q_u) = 1 - A_q A_k, \quad N_q N_k = \frac{1}{|\Omega_N|} \sum_{u \in \Omega_N} s_{\mathrm{norm}}(q_u) = 1 - N_q A_k. \quad (4)$$

Here, for example, $N_q A_k$ is precisely the mean anomaly similarity score $(\overline{s_{\mathrm{anom}}(q)})$ of all normal queries. Across datasets we consistently observe $N_q N_k > A_q N_k$ and $A_q A_k > N_q A_k$ (Figure 2(a)), indicating that similarity score $s_{\mathrm{anom}}(q)$ is a stable anomaly discriminative signal. Motivated by this observation, we replace parametric fitting with similarity retrieval, as adopted in MRAD. Furthermore, as described in section 3.2.3, we introduce a lightweight fine-tuning strategy to further enlarge the $A_q A_k$–$N_q A_k$ gap. (Figure 2(b)). More details see Appendix C.1.

### 3.1.2 APPROACH OVERVIEW

As shown in Figure 3, given an auxiliary dataset, MRAD-TF freezes the CLIP image encoder to extract class and patch tokens from the auxiliary dataset, building a two-level feature-label memory bank. During inference, a query image retrieves against this memory to produce an image-level anomaly score and a pixel-level anomaly map. To further improve discriminability with little cost, MRAD-FT adds two linear layers into cross-atten module to fine-tune the retrieval metric while keeping the backbone frozen. Based on MRAD-FT, we propose MRAD-CLIP, which injects region priors into learnable CLIP prompts, improving generalization and anomaly localization beyond conventional prompt-learning methods. In summary, this retrieval-based access to the original data distribution highlights the strong potential of such paradigms for robust anomaly detection.

### 3.2 MRAD: FROM TRAIN-FREE TO FINE-TUNE

### 3.2.1 FEATURE–LABEL MEMORY BANK

Observing that retrieval against normal/abnormal features is strongly discriminative, we replace trainable explicit classifiers with a two-level feature–label memory bank. We split the frozen CLIP

image encoder into a global branch $\phi_{\mathrm{cls}}$ (emitting the class token) and a local branch $\phi_{\mathrm{vv}}$ (V-V attention, emitting patch tokens), which share the same pretrained weights. For the $i$-th image $I_i$, the global feature and the $u$-th patch feature are

$$k_i^{\mathrm{cls}} = \phi_{\mathrm{cls}}(I_i) \in \mathbb{R}^d, \qquad f_{i,u} = \phi_{\mathrm{vv}}^{(u)}(I_i) \in \mathbb{R}^d, \tag{5}$$

where features are $\ell_2$-normalized for stable retrieval. Given a binary mask label, we downsample it to the patch grid of $\phi_{\mathrm{vv}}$ and use $\tilde{M}_{i,u}$ to denote the label at the $u$-th patch.

For an anomalous image, we obtain two region prototypes by averaging patch features inside and outside the annotated abnormal region, respectively. We denote these patch-level prototypes as $\mu_i^{\mathrm{anom}}$ (only patches with $\tilde{M}_{i,u} = 1$) and $\mu_i^{\mathrm{norm}}$ (only patches with $\tilde{M}_{i,u} = 0$). For a normal-labeled image, we only compute the normal prototype $\mu_i^{\mathrm{norm}}$ (equivalently, $\tilde{M}_i \equiv 0$). Let $e_{\mathrm{norm}} = [1,0]^\top$ and $e_{\mathrm{anom}} = [0,1]^\top$ denote one-hot labels. The memory bank thus has two levels:

(i) an *image-level* memory that stores each global feature $k_i^{\mathrm{cls}}$ with its normal/abnormal label $e_{\mathrm{norm}}$ and $e_{\mathrm{anom}}$;

(ii) a *patch-level* memory that stores the region prototypes $\mu_i^{\mathrm{norm}}$ and $\mu_i^{\mathrm{anom}}$ together with their corresponding labels $e_{\mathrm{norm}}$ and $e_{\mathrm{anom}}$.

Finally, stacking keys and values yields the matrix form

$$K_{\mathrm{cls}} \in \mathbb{R}^{N_c \times d}, \quad V_{\mathrm{cls}} \in \mathbb{R}^{N_c \times 2}, \quad K_{\mathrm{pat}} \in \mathbb{R}^{N_p \times d}, \quad V_{\mathrm{pat}} \in \mathbb{R}^{N_p \times 2}, \tag{6}$$

where $N_c$ is the number of image-level entries and $N_p > N_c$ is the number of patch-level entries.

### 3.2.2 MRAD-TF: TRAIN-FREE BASE MODEL

For a query image $I$, we obtain global and patch queries $Q_{\mathrm{cls}}, Q_{\mathrm{pat}} = \phi(I) \in \mathbb{R}^d, \mathbb{R}^{u \times d}$. The retrieval-based detection process in cross-atten module is defined as:

$$Y_{\mathrm{cls}}^{n/a} = \left[ \mathrm{softmax}\!\left( Q_{\mathrm{cls}} K_{\mathrm{cls}}^\top / \tau \right) V_{\mathrm{cls}} \right]_{n/a}, \qquad Y_{\mathrm{cls}} \in \mathbb{R}^{1 \times 2}, \tag{7}$$

$$Y_{\mathrm{seg}}^{n/a} = \left[ \mathrm{softmax}\!\left( Q_{\mathrm{pat}} K_{\mathrm{pat}}^\top / \tau \right) V_{\mathrm{pat}} \right]_{n/a}, \qquad Y_{\mathrm{seg}} \in \mathbb{R}^{u \times 2}. \tag{8}$$

where $[\cdot]_{n/a}$ selects the `normal`/`anomaly` channel. Here $Y_{\mathrm{seg}}$ represents segmentation logits and denotes the per-patch normal/anomalous score. Note that $Y_{\mathrm{seg}}^a$ corresponds to the anomaly similarity score $s_{\mathrm{anom}}(q)$ introduced in section 3.1.1. For pixel-level segmentation, $Y_{\mathrm{seg}}^a$ is upsampled to form the anomaly map $\hat{M}$. For image-level classification, the final anomaly score is computed as:

$$S(I) = Y_{\mathrm{cls}}^a + \mathrm{TopKMean}(\hat{M}), \tag{9}$$

We define $\mathrm{TopKMean}(\hat{M})$ as the average of the top-$k$ largest values in the upsampled anomaly map $\hat{M}$, which emphasizes high-confidence anomalous regions while suppressing background noise.

### 3.2.3 MRAD-FT: LIGHTWEIGHT FINE-TUNING

As shown in Figure 2, under the train-free setting, the discriminative margins between normal and abnormal similarities are limited. We therefore apply a lightweight fine-tuning that calibrates the retrieval metric to better align normal and abnormal semantics. As a result, the dataset-level statistics (e.g., $\mathrm{A}_q\mathrm{A}_k - \mathrm{N}_q\mathrm{A}_k$) exhibit larger separations, leading to more reliable anomaly decisions. Specifically, for query-key pairs, we insert two linear mappings $W_q, W_k \in \mathbb{R}^{d \times d}$, replacing the $QK^\top$ in section 3.2.2 with $(QW_q)(KW_k)^\top$. The fine-tuned retrieval process is defined as:

$$Y_{\mathrm{cls}}^{n/a} = \left[ \mathrm{softmax}\!\left( \frac{(Q_{\mathrm{cls}} W_{q\_\mathrm{cls}})(K_{\mathrm{cls}} W_{k\_\mathrm{cls}})^\top}{\tau} + \mathcal{M}_\rho(Q_{\mathrm{cls}}, K_{\mathrm{cls}}) \right) V_{\mathrm{cls}} \right]_{n/a}, \tag{10}$$

$$Y_{\mathrm{seg}}^{n/a} = \left[ \mathrm{softmax}\!\left( \frac{(Q_{\mathrm{pat}} W_{q\_\mathrm{seg}})(K_{\mathrm{pat}} W_{k\_\mathrm{seg}})^\top}{\tau} + \mathcal{M}_\rho(Q_{\mathrm{pat}}, K_{\mathrm{pat}}) \right) V_{\mathrm{pat}} \right]_{n/a}. \tag{11}$$

To stabilize training, we apply a similarity dropout operator $\mathcal{M}_\rho(\cdot, \cdot)$ that masks the top-$\rho\%$ highest similarities in the $QK^\top$ matrix for each query, setting their logits to $-\infty$ during training (retaining all matches during inference). This prevents trivial self-matching in single-dataset settings and encourages robust discrimination on less similar features. Given the classification label y and segmentation mask $M$, the objective function combines classification and segmentation losses:

$$\mathcal{L} = \mathcal{L}_{\text{cls}} + \mathcal{L}_{\text{seg}} = \text{BCE}(Y_{\text{cls}}, y) + \text{Dice}(Y_{\text{seg}}, M) + \text{Focal}(Y_{\text{seg}}, M), \tag{12}$$

where $\text{BCE}(\cdot, \cdot)$ denotes a binary cross-entropy loss, $\text{Dice}(\cdot, \cdot)$ denotes a dice loss (Li et al., 2019), and $\text{Focal}(\cdot, \cdot)$ denotes a focal loss (Lin et al., 2017).

### 3.3 MRAD-CLIP: DYNAMIC PROMPT-LEARNING EXTENSION

To leverage language priors while keeping the vision branch frozen, we augment the CLIP prompt with learnable context tokens and inject image-specific priors from MRAD-FT.

**Static, object-agnostic prompts.** We prepend $E$ learnable context tokens to the textual templates "good object" and "damaged object" to form the class-agnostic prompts:

$$\begin{aligned} P^n &= [V_1^n][V_2^n] \cdots [V_E^n][\text{good object}], \\ P^a &= [V_1^a][V_2^a] \cdots [V_E^a][\text{damaged object}]. \end{aligned} \tag{13}$$

**Prior-guided dynamic bias.** Given a query image, we threshold the anomaly map from MRAD-FT to separate anomalous and normal regions, and average the patch features in each region to obtain prototypes. These prototypes are projected through lightweight linear layers into bias vectors $b^{n/a}$, which are then added to the $E$ context tokens to form the image-specific dynamic prompts $P_{\text{dyn}}^{n/a}$.

$$\begin{aligned} P_{\text{dyn}}^n &= [V_1^n + b^n]\,[V_2^n + b^n]\,\cdots\,[V_E^n + b^n][\text{good object}], \\ P_{\text{dyn}}^a &= [V_1^a + b^a]\,[V_2^a + b^a]\,\cdots\,[V_E^a + b^a][\text{damaged object}], \end{aligned} \tag{14}$$

**Prompt-conditioned cosine classifier.** Given the CLIP text encoder $T(\cdot)$ (following the same setting as AnomalyCLIP) and $\ell_2$ normalization $\text{Norm}(\cdot)$, we obtain the normalized text embeddings $t_{\text{norm}} = \text{Norm}\big(T(P_{\text{dyn}}^n)\big)$ and $t_{\text{anom}} = \text{Norm}\big(T(P_{\text{dyn}}^a)\big)$ from the CLIP text encoder. Then we use the prompt-conditioned cosine classifier to predict:

$$Y_{\text{seg}}^{n/a} = \text{softmax}\left(\frac{1}{\tau}\big[\,\cos(t_{\text{norm}}, Q_{\text{pat}}),\ \cos(t_{\text{anom}}, Q_{\text{pat}})\,\big]\right) \in \mathbb{R}^{u \times 2}, \tag{15}$$

where $\cos(t, Q_{\text{pat}})$ returns the vector of cosine similarities between $t$ and all patch features. The refined anomaly map is then obtained by upsampling $Y_{\text{seg}}^a$ to the original image resolution and the image-level anomaly score follows Equation 9. Only the text-side parameters are trained while the vision encoder and the MRAD memories remain frozen. We use the same losses as in Equation 12 for classification and segmentation, resulting in a prompt-conditioned refinement that improves local alignment and reduces background false positives with negligible extra parameters.

## 4 EXPERIMENTS

### 4.1 EXPERIMENTAL SETUP

**Datasets.** We evaluate the ZSAD performance on 16 public datasets, covering both industrial and medical domains. In the industrial domain, we adopt eight widely used datasets: MVTec-AD (Bergmann et al., 2019), VisA (Zou et al., 2022), BTAD (Mishra et al., 2021), MPDD (Jezek et al., 2021), SDD (Jezek et al., 2021), KSDD2 (Božič et al., 2021), DAGM (Wieler & Hahn, 2007), and DTD-Synthetic (Aota et al., 2023). In the medical domain, we include eight datasets: HeadCT (Salehi et al., 2021), BrainMRI (Kanade & Gumaste, 2015), Br35H (Hamada., 2020), CVC-ColonDB (Tajbakhsh et al., 2015), CVC-ClinicDB (Bernal et al., 2015), Endo (Hicks et al., 2021), Kvasir (Jha et al., 2019), and ISIC (Codella et al., 2018). Following the ZSAD setting, we use a single industrial dataset as the auxiliary training source and directly test on the remaining industrial and all medical datasets. By default, VisA is adopted as the auxiliary dataset to build the memory bank, since its categories are disjoint from the others; when evaluating VisA itself, we instead use MVTec-AD as the auxiliary dataset. More details see Appendix B.

Table 1: ZSAD results on 16 datasets. Top: pixel-level (P-AUROC% / PRO%). Bottom: image-level (I-AUROC% / I-AP%). Higher is better. Red and Blue denote the best and second-best *training-based* results per dataset/metric; Orange highlights better results under *train-free* setting.

(a) Pixel-level ZSAD (P-AUROC% / PRO%).

| Method → | TRAIN-FREE METHODS | | TRAINING-BASED METHODS | | | | |
|---|---|---|---|---|---|---|---|
| Dataset ↓ | WinCLIP (CVPR'23) | MRAD-TF (Ours) | AdaCLIP (ECCV'24) | AnomalyCLIP (ICLR'24) | FAPrompt (ICCV'25) | MRAD-FT (Ours) | MRAD-CLIP (Ours) |
| *Industrial* | | | | | | | |
| MVTec-AD | (85.1, 64.6) | (86.7, 63.5) | (86.8, 33.8) | (91.1, 81.4) | (90.6, 83.3) | (92.2, 85.4) | (93.0, 86.8) |
| VisA | (79.6, 56.8) | (91.0, 71.0) | (95.1, 71.3) | (95.5, 87.0) | (95.9, 87.5) | (95.9, 89.1) | (95.9, 88.0) |
| BTAD | (71.4, 32.8) | (80.5, 36.3) | (87.7, 17.1) | (93.3, 69.3) | (91.7, 69.0) | (94.7, 74.3) | (95.4, 72.8) |
| MPDD | (71.2, 48.9) | (92.7, 76.2) | (95.2, 10.8) | (96.2, 79.7) | (96.7, 75.9) | (97.4, 90.6) | (97.9, 90.6) |
| DTD-Synthetic | (78.4, 51.0) | (90.7, 73.8) | (94.1, 24.9) | (97.6, 88.3) | (97.7, 89.4) | (97.2, 90.8) | (98.1, 89.8) |
| SDD | (55.9, 14.7) | (85.0, 63.0) | (79.5, 4.9) | (90.1, 62.9) | (89.3, 63.9) | (91.0, 71.2) | (93.0, 72.0) |
| KSDD2 | (75.4, 69.2) | (94.9, 87.1) | (85.8, 72.9) | (97.9, 94.9) | (97.4, 93.2) | (98.8, 89.8) | (98.9, 95.6) |
| DAGM | (75.5, 44.4) | (86.9, 66.1) | (76.2, 56.3) | (96.5, 88.4) | (95.6, 89.1) | (96.1, 74.9) | (97.4, 90.3) |
| *Medical* | | | | | | | |
| ISIC | (83.5, 55.1) | (84.4, 72.1) | (85.4, 5.3) | (88.4, 78.1) | (88.9, 81.2) | (90.9, 83.6) | (91.3, 83.4) |
| CVC-ColonDB | (64.8, 28.4) | (77.6, 59.6) | (79.3, 6.5) | (81.9, 71.4) | (82.5, 70.7) | (82.8, 72.9) | (84.7, 73.9) |
| CVC-ClinicDB | (70.7, 32.5) | (81.6, 59.0) | (84.3, 14.6) | (85.9, 69.6) | (84.7, 68.1) | (85.9, 72.2) | (87.3, 73.9) |
| Kvasir | (69.8, 31.0) | (77.1, 52.3) | (79.4, 12.3) | (81.9, 45.7) | (82.0, 43.9) | (83.9, 52.7) | (84.3, 52.7) |
| Endo | (68.2, 28.3) | (82.7, 60.0) | (84.0, 10.5) | (86.3, 67.3) | (86.5, 67.2) | (87.3, 70.0) | (88.3, 71.6) |
| **Average** | (73.0, 42.9) | (85.5, 64.6) | (85.6, 26.2) | (91.0, 75.7) | (90.7, 75.6) | (91.9, 78.3) | (92.7, 80.1) |

(b) Image-level ZSAD (I-AUROC% / I-AP%).

| Method → | TRAIN-FREE METHODS | | TRAINING-BASED METHODS | | | | |
|---|---|---|---|---|---|---|---|
| Dataset ↓ | WinCLIP (CVPR'23) | MRAD-TF (Ours) | AdaCLIP (ECCV'24) | AnomalyCLIP (ICLR'24) | FAPrompt (ICCV'25) | MRAD-FT (Ours) | MRAD-CLIP (Ours) |
| *Industrial* | | | | | | | |
| MVTec-AD | (91.8, 95.1) | (79.0, 90.0) | (92.0, 96.4) | (91.5, 96.2) | (91.9, 95.7) | (92.3, 96.6) | (94.0, 97.4) |
| VisA | (78.1, 77.5) | (75.0, 79.1) | (83.0, 84.9) | (82.1, 85.4) | (84.5, 86.8) | (85.5, 87.8) | (85.7, 88.3) |
| BTAD | (83.3, 84.1) | (86.3, 90.3) | (91.6, 92.4) | (89.1, 91.1) | (91.2, 89.1) | (94.5, 97.7) | (92.8, 94.2) |
| MPDD | (61.5, 69.2) | (67.9, 73.8) | (76.4, 80.4) | (73.7, 76.5) | (76.9, 85.9) | (79.9, 80.6) | (81.8, 83.4) |
| DTD-Synthetic | (63.8, 83.5) | (79.1, 89.6) | (92.8, 97.0) | (94.5, 97.7) | (95.4, 97.8) | (94.1, 97.8) | (96.0, 98.4) |
| SDD | (54.8, 37.7) | (80.0, 66.3) | (77.5, 65.7) | (82.7, 73.8) | (83.4, 75.3) | (83.8, 75.8) | (83.9, 76.2) |
| KSDD2 | (77.5, 43.9) | (90.4, 82.6) | (90.5, 77.7) | (90.8, 76.5) | (94.4, 86.9) | (92.9, 86.6) | (95.1, 88.9) |
| DAGM | (58.5, 59.1) | (78.7, 80.6) | (77.6, 72.7) | (98.0, 97.7) | (98.0, 98.1) | (98.0, 98.1) | (98.4, 98.6) |
| *Medical* | | | | | | | |
| HeadCT | (83.7, 81.6) | (69.3, 73.6) | (93.4, 92.2) | (95.3, 95.2) | (96.0, 96.4) | (96.2, 96.9) | (97.1, 97.6) |
| BrainMRI | (92.0, 90.7) | (94.1, 96.8) | (94.9, 94.2) | (96.1, 92.3) | (95.7, 95.6) | (96.9, 97.0) | (97.0, 97.4) |
| Br35H | (80.5, 82.2) | (91.0, 92.1) | (95.7, 95.7) | (97.3, 96.1) | (97.0, 95.4) | (97.7, 96.5) | (97.9, 97.6) |
| **Average** | (75.1, 73.2) | (81.0, 83.2) | (87.8, 86.3) | (90.1, 88.9) | (91.3, 91.2) | (92.0, 91.9) | (92.7, 92.5) |

**Evaluation metrics.** For image-level classification tasks, we report Area Under the Receiver Operating Characteristic Curve (AUROC) and Average Precision (AP). For segmentation tasks, we adopt pixel-level AUROC and the Per-Region Overlap (PRO) metric to assess localization performance.

**Implementation details.** We follow prior work and use the publicly available CLIP (ViT-L/14-336) pretrained weights, with input resolution unified to $518 \times 518$. We extract both class token and patch tokens from the last layer of the image encoder to construct the memory bank: when using MVTec-AD as the auxiliary dataset, the memory bank contains 1725/2976 (class/patch) entries; when using VisA, it contains 2162/3093 entries. For MRAD-CLIP, we set the learnable prompt length to 12. We train with the Adam (Kingma & Ba, 2014) optimizer, using a learning rate of $5 \times 10^{-4}$ and batch size of 8. MRAD-FT is trained for only 1 epoch, followed by MRAD-CLIP for 5 epochs. During training, the vision backbone is frozen and only our lightweight modules are updated. All experiments are conducted on a single NVIDIA RTX 3090 (24GB). We report the averaged results across all categories of each target dataset. More details see Appendix A.1.

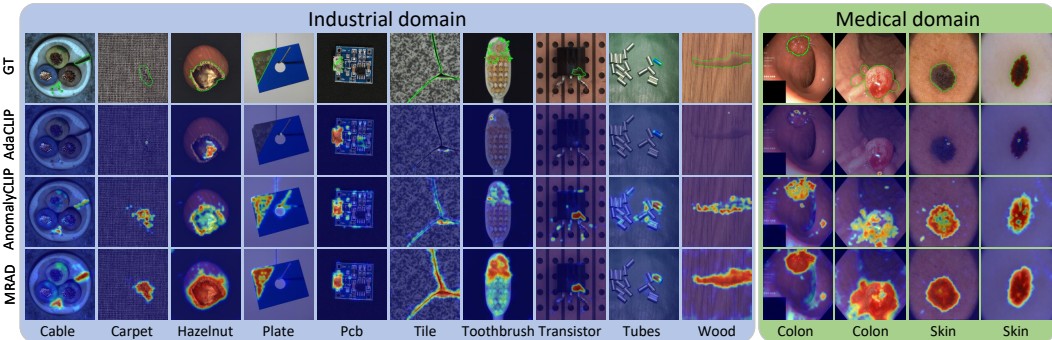

Figure 4: Comparison of anomaly segmentation results between MRAD-CLIP and other methods.

## 4.2 COMPARISON WITH STATE-OF-THE-ART METHODS

**Quantitative Comparison.** As shown in Table 1, we report results on 16 datasets spanning industrial and medical domains. MRAD-TF already surpasses prior train-free baselines (e.g., WinCLIP), indicating that direct access to the empirical feature-label distribution is highly effective. Building on this train-free model, the MRAD-FT further outperforms training-based methods (e.g., FAPrompt, AnomalyCLIP) on both pixel-level and image-level metrics. Further, MRAD-CLIP attains state-of-the-art results for ZSAD with consistent margins over the strongest prior work (including our MRAD-FT), indicating that dynamic prompts conditioned on region priors strengthen cross-domain generalization and sharpen localization. Taken together, these results demonstrate that our MRAD is a strong ZSAD method, and can be easily extended in a principled manner to achieve state-of-the-art performance, underscoring the effectiveness and flexibility of the framework. We provide more detailed analyses of the mechanism behind MRAD's cross-class retrieval generalization in the Appendix A.2 and C.2.

**Qualitative Comparison.** In industrial datasets (diverse materials, textures, and imaging conditions), MRAD-FT and MRAD-CLIP consistently rank among the top on both image-level and pixel-level metrics. They also exhibit low across-dataset variance and minimal rank fluctuations, indicating robustness to background clutter and fine-grained defects. In medical datasets (large morphological variability and strong domain shift), the same pattern holds: the lightweight metric calibration and prior-guided prompting preserve decision margins and localization quality under cross-domain transfer. Figure 4 visualizes anomaly maps from both industrial and medical datasets. Qualitatively, the resulting heatmaps show fewer background false positives, tighter anomaly boundaries, and smoother localization of large defective regions across domains. These observations mirror the quantitative gains and further support the method's stability and generalization. A comparison of parameter counts and inference time with other methods is provided in the Appendix C.3. More visualization results are provided in Appendix D.

## 4.3 ABLATION STUDY

We conduct extensive ablation studies to examine the contributions of different components in the MRAD framework. All results are reported on representative industrial or medical datasets. The main findings are summarized below, while additional ablations and more details are provided in Appendix C.4 for completeness.

**Ablation on TF/FT/CLIP.** Across six industrial datasets, as shown in Figure 5, the radar plots reveal a clear monotonic trend. MRAD-TF establishes competitive performance, while adding two linear layers in MRAD-FT yields consistent gains on both image-level (I-AUROC/I-AP) and pixel-level (P-AUROC/PRO) metrics. This indicates that lightweight fine-tuning effectively enhances the discriminability of classification and segmentation. Based on this, MRAD-CLIP further improves especially the pixel-level metrics by dynamic prompt learning with region priors, while also yielding small but consistent gains on image-level scores. The outward expansion of the curves across all axes confirms that each component contributes complementary benefits, delivering stable improvements rather than dataset-specific spikes.

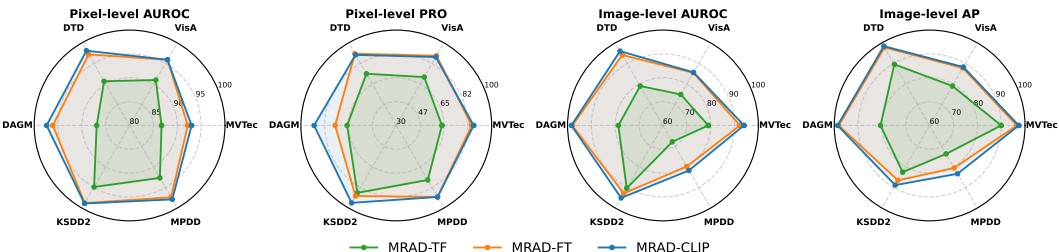

Figure 5: Ablation results of different MRAD variants. Radar charts on six industrial datasets compare image-level (I-AUROC/I-AP) and pixel-level (P-AUROC/PRO) metrics.

Table 2: Ablation on dynamic prompt biases. Results are reported as Pixel-level (AUROC, PRO) and Image-level (AUROC, AP). The best performance in each column is highlighted in bold.

| bias ($b^{n/a}$) | MVTec-AD | | VisA | | Kvasir | HeadCT |
|---|---|---|---|---|---|---|
| | Pixel | Image | Pixel | Image | Pixel | Image |
| static (**baseline**) | (91.5, 82.9) | (92.0, 96.3) | (95.6, 87.3) | (85.4, 87.9) | (83.5, 49.5) | (96.1, 97.0) |
| + class token | (86.2, 54.9) | (89.6, 95.1) | (95.1, 86.8) | (84.9, 87.7) | (79.0, 45.2) | (90.4, 90.8) |
| + cross-patch | (92.5, 85.7) | (93.0, 96.9) | (95.3, **88.6**) | (85.2, **88.4**) | (83.0, 50.0) | (**97.4, 97.6**) |
| + anomaly prior | (92.8, 85.2) | (93.6, 97.3) | (95.6, 88.1) | (85.6, 88.2) | (83.7, 51.8) | (96.5, 96.7) |
| + dual prior | (**93.0, 86.8**) | (**94.0, 97.4**) | (**95.9**, 88.0) | (**85.7**, 88.3) | (**84.3, 52.7**) | (97.1, **97.6**) |

**Ablation on dynamic prompt biases.** We change only the source of the additive bias in MRAD-CLIP's dynamic prompts and keep all other components fixed. Table 2 reports pixel-level (AUROC/PRO) and image-level (AUROC/AP) results on four datasets. The *static* variant (no bias; equivalent to AnomalyCLIP) serves as the baseline. Bias from the *class token* (similar to CoCoOp) degrades segmentation and cross-domain robustness, indicating that global features alone are insufficient. Introducing *cross-patch* attention (two learnable queries attending to patch features) improves ZSAD performance, suggesting that conditioning prompts on region-level context helps generalization to unseen categories. Guided by this observation, conditioning on the MRAD-FT *anomaly prior* yields consistent image-level and pixel-level gains. Furthermore, injecting both normal and anomalous priors (*dual prior*) achieves nearly the best and most robust performance across datasets. These results indicate that local region-aware biases with explicit priors are crucial for ZSAD, while global-only features tend to weaken performance. More details see Appendix C.4.

**Memory size ablation.** We ablate the patch-level memory by random subsampling to $n \in \{\text{Full}, 2000, 1000, 500, 100\}$. For each $n$, we average across multiple random subsamplings and report pixel AUROC on MVTec-AD for MRAD-FT and MRAD-CLIP (Figure 6). As $n$ decreases, both methods show a smooth but minor decline, due to the reduced prototype coverage in the memory bank. However, the drop is only within a few tenths, demonstrating the strong robustness of the MRAD framework. Relatively, MRAD-CLIP is more robust even under small memory sizes, exhibiting smaller drops and lower variance, primarily because it does not rely on precise dynamic biases from prompts and can already benefit from coarse regional priors. In practice, we believe the diversity-aware memory even with a small-size memory ($n \geq 100$) can track near-full performance under typical budgets.

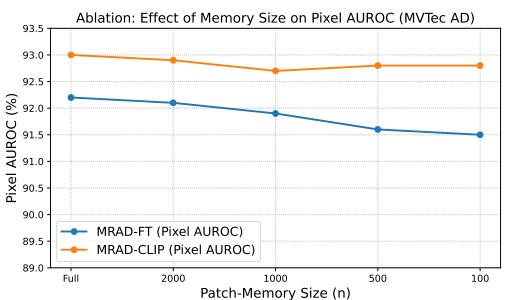

Figure 6: Metrics about the ablation of Memory size.

## 5 CONCLUSION

In this paper, we explore an alternative to mainstream parametric fitting for anomaly detection by directly exploiting the empirical distribution of auxiliary data. Building on this idea, we propose

MRAD as a unified memory-driven retrieval framework. The base model, MRAD-TF, constructs a two-level memory bank on top of a frozen vision backbone and solves both classification and segmentation via similarity retrieval. Further, we develop two lightweight variants, MRAD-FT and MRAD-CLIP. Experiments across diverse industrial and medical benchmarks show that all variants exhibit strong zero-shot capability and robust performance under cross-domain evaluation. We believe this paradigm provides a simple and robust baseline for ZSAD, and it lays the foundation for scaling to larger datasets and online or incremental settings.

## ACKNOWLEDGMENTS

This work is supported by the National Science and Technology Major Project (2022ZD0119400), the National Natural Science Foundation of China (Grant Nos. 62303458 and 62303461), and the Beijing Municipal Natural Science Foundation (China) (Grant No. 4252053).

## REPRODUCIBILITY STATEMENT

To facilitate reproducibility and completeness, we include an Appendix composed of five sections. In Appendix A, we provide implementation details of MRAD, including model components, training/inference recipes, full hyperparameter settings, with additional mechanism explanations and retrieval visualizations. Appendix B describes the key statistics of datasets used in our experiments and evaluation protocols under the ZSAD setting. Appendix C reports additional experimental results, including motivation studies, ablation analyses, as well as evaluations under a broader range of experimental settings. Appendix D presents extended qualitative analyses and visualizations of our method. In addition, the usage details of LLMs are provided in Appendix E.

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

Appendix

# A  Implementation Details And Baselines

## A.1  Implementation Details

We follow prior work and use the publicly available CLIP (ViT-L/14-336) pretrained weights, with input resolution unified to $518 \times 518$. We extract both class token and patch tokens from the last layer of the image encoder to construct the memory bank: when using MVTec-AD as the auxiliary dataset, the memory bank contains $1725/2976$ (class/patch) entries; when using VisA, it contains $2162/3093$ entries.

For MRAD-CLIP, we set the length of the learnable prompt to 12 and set the replacing token number to 4 in text encoder. Training is performed with the Adam optimizer (Kingma & Ba, 2014), using a learning rate of $5 \times 10^{-4}$ and a batch size of 8. MRAD-FT is trained for 1 epoch, followed by MRAD-CLIP for 5 epochs.

For cross-dataset evaluation, when testing on MVTec-AD, training and memory bank construction are performed on VisA, and vice versa. Unless otherwise specified, the batch size is set to 8, and the retrieval temperature coefficient $\tau$ is fixed to 1. For computing image-level anomaly scores, we use the top-$k$ mean aggregation in Equation 9, where $k$ is set to 1% of the total number of pixels in the anomaly map. In MRAD-FT, we set similarity-masking threshold $\rho_{\text{seg}} = 20\%$ for segmentation and $\rho_{\text{cls}} = 5\%$ for classification. In MRAD-CLIP, the threshold for generating anomaly priors is set to 0.5 by default. All experiments are conducted on a single NVIDIA RTX 3090 GPU (24GB). We report averaged results across all categories of each target dataset.

## A.2  Mechanism Behind Cross-Class Retrieval Generalization in MRAD

We reformulate zero-shot anomaly detection (ZSAD) as a retrieval problem over a feature–label memory, instead of compressing the entire auxiliary set (e.g., VisA) into a parametric classifier head. All features and labels from the auxiliary dataset are preserved verbatim in the memory bank, allowing query samples from other datasets to fully retrieve and compare against these prototypes. Because every stored prototype participates in the final similarity computation, this design avoids the information-loss issues inherent to collapsing a rich empirical distribution into a small set of classifier weights.

During training, each sample is required to retrieve not only its own category but also prototype clusters from all other categories in the memory. In other words, the query representation is always matched against a global, multi-class memory bank. This training procedure effectively simulates the deployment scenario where the model must perform cross-class, cross-scenario retrieval, forcing it to learn category-agnostic anomaly patterns rather than depending on category-specific textures or contextual correlations.

At inference time, new unseen categories simply act as new query samples. They are evaluated against the same fixed memory, using exactly the same retrieval mechanism as during training. Consequently, the retrieved prototypes remain aligned with the learned, category-agnostic anomaly representation. This yields stronger robustness and more reliable generalization both across classes and across datasets.

## A.3  More details about MRAD modules

### A.3.1  Details about Image Encoder

In our framework, the memory bank is constructed directly from the frozen CLIP image encoder. Specifically, we adopt the frozen Vision Transformer (ViT) (Dosovitskiy et al., 2020) backbone of CLIP, which encodes each input image into a sequence of patch embeddings along with a class token. Standard Q-K attention in the ViT encoder often focuses heavily on a few dominant tokens, which benefits global object recognition but may suppress fine-grained local semantics. For anomaly segmentation, however, local context is critical. To mitigate this bias, we adopt a V-V attention mechanism to all ViT layers inspired by prior work. By default, we employ V-V attention to

obtain patch features for memory-bank construction, training, and inference, while the original Q-K attention is retained to generate global features.

Concretely, unlike standard self-attention where both $Q$ and $K$ are projected from the same input features, V-V attention replaces them entirely with the value embeddings $V$. The attention thus takes the form

$$\text{Output} = \text{Softmax}\left(\frac{VV^\top}{\sqrt{d}}\right) V, \tag{16}$$

which enforces interactions purely among value tokens and encourages each token to attend primarily to its local context. This design mitigates the dominance of a single strong embedding and suppresses bias from global object tokens. As a result, the attention map exhibits a more diagonal structure, encouraging each token to focus primarily on its corresponding local context rather than being distracted by distant high-activation regions.

In practice, this modification preserves local visual semantics in the feature representation, leading to tighter anomaly boundaries and fewer background false positives. We provide only a brief overview here and refer readers to Zhou et al. (2024) for the complete derivation.

### A.3.2 DETAILS ABOUT TEXT ENCODER

To refine the textual space in MRAD-CLIP, we follow prior CLIP-based ZSAD works (Zhou et al., 2024; Cao et al., 2024; Qu et al., 2024) and introduce a lightweight prompt tuning strategy. The CLIP text encoder remains frozen, and we insert a small set of randomly initialized learnable prompt tokens starting from a lower layer and moving upward. At each Transformer block, these tokens are concatenated with the original token sequence so that self-attention can mix them with the textual features. After the block, the used prompts are discarded and a fresh set is initialized for the next layer. This step-by-step design provides gradual calibration and prevents prompt drift. Throughout training, only the prompts are updated, while the backbone weights remain fixed. This progressively refines the textual space to better encode normal and abnormal semantics with very few trainable parameters. The specific parameter settings follow AnomalyCLIP (Zhou et al., 2024).

### A.4 BASELINES AND EXPERIMENTAL PROTOCOL

To ensure fairness, we follow a unified experimental protocol: VisA is used as the auxiliary dataset for training by default, and evaluation is conducted on all other datasets. Baselines and reproduction details are given as follows.

**WinCLIP (Jeong et al., 2023)** is the first work to employ frozen CLIP for ZSAD. The training-free approach employs window/patch sampling and computes text-image similarity at the region level to localize anomalies. Anomaly scores are aggregated from the dissimilarity between visual patches and a textual description of normality. As the official implementation of WinCLIP is unavailable, we adopt the reproduced code for re-implementation under our unified protocol.

**AdaCLIP (Cao et al., 2024)** introduces learnable prompts for CLIP and optimizes them on auxiliary anomaly-detection data. It proposes *static* prompts (shared across images) and *dynamic* prompts (generated per test image), as well as their hybrid, to adapt CLIP to ZSAD. We follow the authors' settings where applicable, while using the official code to retrain it with our protocol.

**AnomalyCLIP (Zhou et al., 2024)** learns object-agnostic text prompts that capture generic normality and abnormality, encouraging the model to focus on abnormal regions instead of category semantics. This design improves cross-domain generalization for ZSAD. We follow the authors' settings where applicable, while using the official code to retrain it with our protocol.

**FAPrompt (Zhu et al., 2025)** targets fine-grained ZSAD by learning *abnormality prompts* formed by a compound of shared normal tokens and a few learnable abnormal tokens. It further introduces a data-dependent abnormality prior to produce dynamic learning prompts on each test image. We follow the authors' settings where applicable, while using the official code to retrain it with our protocol.

Table 3: Key statistics of the 16 industrial and medical datasets used in our study.

| Domain | Dataset | Classes | Normal Samples | Anomaly Samples | Data type | Mask Labels |
|---|---|---|---|---|---|---|
| Industrial | MVTec-AD | 15 | 467 | 1258 | object & texture | ✓ |
| | VisA | 12 | 962 | 1200 | object | ✓ |
| | BTAD | 3 | 451 | 290 | object & texture | ✓ |
| | MPDD | 6 | 176 | 282 | object | ✓ |
| | DTD-Synthetic | 12 | 357 | 947 | texture | ✓ |
| | SDD | 1 | 181 | 74 | object | ✓ |
| | KSDD2 | 1 | 894 | 110 | object | ✓ |
| | DAGM | 10 | 2000 | 2000 | texture | ✓ |
| Medical | HeadCT | 1 | 100 | 100 | brain | × |
| | BrainMRI | 1 | 98 | 155 | brain | × |
| | Br35H | 1 | 1500 | 1500 | brain | × |
| | ISIC | 1 | 0 | 379 | skin | ✓ |
| | CVC-ColonDB | 1 | 0 | 380 | colon | ✓ |
| | CVC-ClinicDB | 1 | 0 | 612 | colon | ✓ |
| | Kvasir | 1 | 0 | 1000 | colon | ✓ |
| | Endo | 1 | 0 | 200 | colon | ✓ |

## B MORE DETAILS ABOUT DATASETS

### B.1 DATASETS

We evaluate the ZSAD performance on 16 public datasets, covering both industrial and medical domains. In the industrial domain, we adopt eight widely used benchmarks: MVTec-AD (Bergmann et al., 2019), VisA (Zou et al., 2022), BTAD (Mishra et al., 2021), MPDD (Jezek et al., 2021), SDD (Jezek et al., 2021), KSDD2 (Božič et al., 2021), DAGM (Wieler & Hahn, 2007), and DTD-Synthetic (Aota et al., 2023). In the medical domain, we include eight datasets: HeadCT (Salehi et al., 2021), BrainMRI (Kanade & Gumaste, 2015), Br35H (Hamada., 2020), CVC-ColonDB (Tajbakhsh et al., 2015), CVC-ClinicDB (Bernal et al., 2015), Endo (Hicks et al., 2021), Kvasir (Jha et al., 2019), and ISIC (Codella et al., 2018). We resize all images and their corresponding segmentation masks to $518 \times 518$ for consistency across datasets. For datasets that do not provide segmentation labels or classification labels, we restrict the evaluation to pixel-level or image-level metrics accordingly. As shown in Table 3, it summarizes the datasets used in our study, including 8 industrial and 8 medical benchmarks. For each dataset, we list the number of categories, normal and anomaly samples, data type, and whether segmentation labels are provided. These benchmarks cover diverse domains, ensuring that our evaluation spans a wide range of anomaly detection scenarios.

## C ADDITIONAL EXPERIMENTAL RESULTS

### C.1 EXTENDED CROSS-DATASET SIMILARITY ANALYSIS

To complement Figure 2 in the main text, we extend the query-key similarity analysis from the four datasets in the main text to *all* benchmarks. Following section 3.1.1, we report the four statistics $A_q A_k$, $A_q N_k$, $N_q A_k$, and $N_q N_k$ under the same protocol. We conduct the cross-dataset study by taking patch features-labels from VisA as keys-values and patch features from other datasets as queries. By definition, $N_q A_k$ is the *average anomaly-similarity* $\overline{s_{\text{anom}}(q)}$ over all *normal* queries, $A_q A_k$ is the average anomaly-similarity over *anomalous* queries, and $N_q N_k / A_q N_k$ are the corresponding averages of the *normal-similarity* $\overline{s_{\text{norm}}(q)}$ for normal/anomalous queries.

Figs. 7–8 extend the query-key similarity study to another eight datasets. In the *train-free* setting (Figure 7), the per-dataset means of the four relations $N_q N_k$, $A_q N_k$, $A_q A_k$, and $N_q A_k$ follow a consistent ordering $N_q N_k > A_q N_k$ and $A_q A_k > N_q A_k$ across datasets, showing that similarity to normal/abnormal keys provides stable discriminative signals without training. This observation is

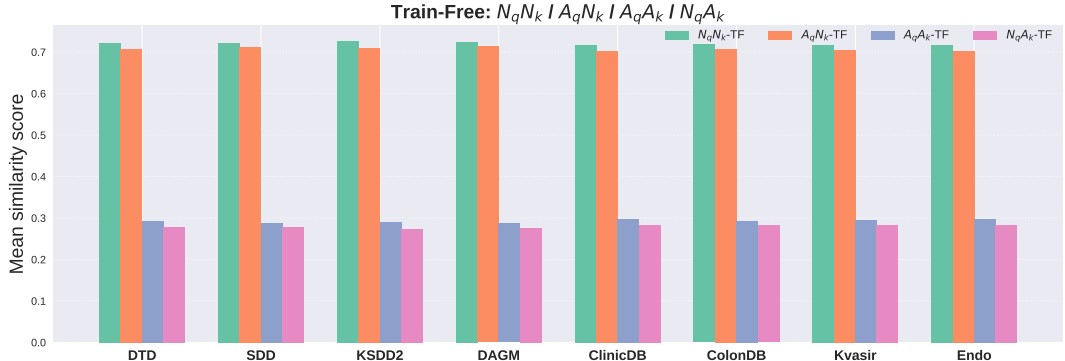

Figure 7: Train-Free per-dataset mean similarity scores across eight datasets.

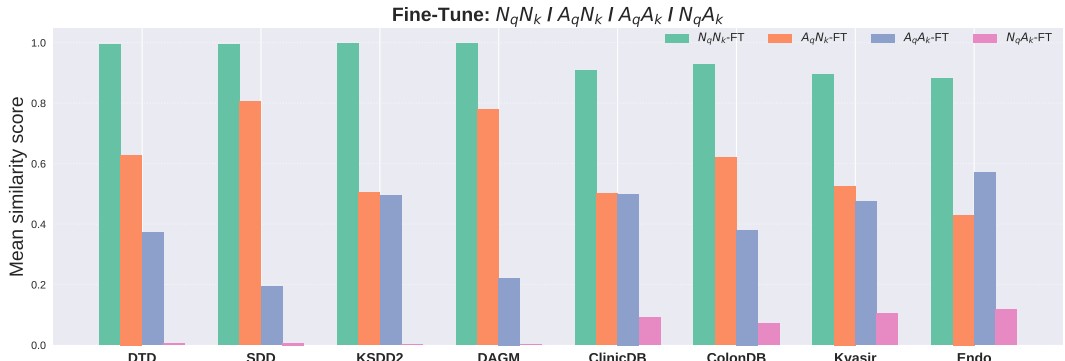

Figure 8: Fine-Tuned per-dataset mean similarity scores across eight datasets.

consistent with the conclusions reported in the main text. After *lightweight fine-tuning* (MRAD-FT; Figure 8), the same ordering is preserved while holding stronger separability. Figure 9 summarizes this effect via the margins $\Delta(A) = A_q A_k - N_q A_k$ and $\Delta(N) = N_q N_k - A_q N_k$: both margins grow substantially on every dataset, confirming that fine-tuning enlarges the gap between positive and negative relations. The central insight of our method is to leverage the discriminative margin as the basis for anomaly detection.

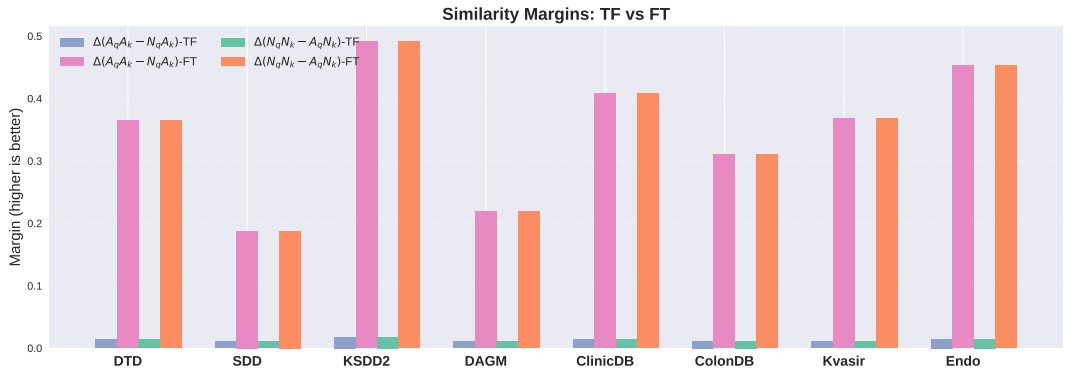

Figure 9: Per-dataset similarity margins under Train-Free and Fine-Tuned settings.

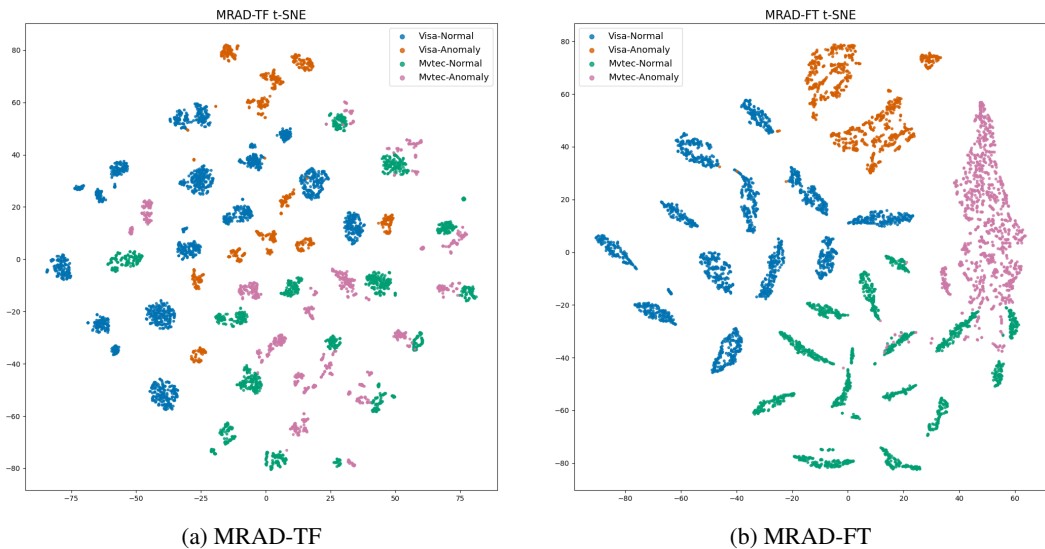

(a) MRAD-TF                                      (b) MRAD-FT

Figure 10: t-SNE visualization of normal and anomalous prototypes from VisA and MVTec.

## C.2 T-SNE VISUALIZATION

To further illustrate the underlying mechanism of our approach, we visualize the feature space of MRAD-TF and MRAD-FT using t-SNE on VisA and MVTec. Fig. 10(a,b) show four types of prototypes: VisA-Normal, VisA-Anomaly, MVTec-Normal, and MVTec-Anomaly.

In the MRAD-TF setting (Fig. 10a), the pretrained CLIP visual encoder induces a feature space in which normal and abnormal prototypes from different categories and datasets tend to separate along a common direction within local mixed-class neighborhoods, even though the global structure is still strongly class-clustered and far from linearly separable by a single hyperplane. This indicates that CLIP not only aligns multiple categories, but also encodes a weak yet consistent "anomaly direction": abnormal prototypes are systematically shifted relative to normal ones in the high-dimensional space. MRAD-TF directly exploits this structure. Instead of learning an additional classifier head on the source domain, it stores normal/abnormal prototypes of all classes into a memory bank and decides whether a query feature has moved along this anomaly direction by cross-class similarity retrieval. Since similarities to all stored prototypes jointly contribute to the final decision, the shared anomaly direction can be reused by target-domain anomalies: their relation to source-domain anomaly prototypes becomes the main decision signal. As a result, even without any fine-tuning, MRAD-TF maintains competitive ZSAD performance on MVTec.

In the MRAD-FT setting (Fig. 10b), we further amplify this implicit anomaly direction and remove its dependence on specific classes. During training, samples from one class are forced to retrieve and learn against memory prototypes from all other classes, effectively simulating "class-shifted" or "dataset-shifted" deployment inside the source domain. In parallel, a lightweight metric calibration via linear Q/K projections aligns queries and memory into a shared subspace, in which the anomaly direction becomes more coherent and salient. The t-SNE visualization shows that, after fine-tuning, abnormal prototypes from different datasets form a tighter, nearly class-agnostic cluster, while normal prototypes are pushed further away and retain only a weak class-cluster structure. In other words, fine-tuning reconstructs the weak anomaly signal that originally lived near class-boundary fringes into an explicit, class-agnostic "universal anomaly pattern." Consequently, when new categories from the target domain are introduced, one can still perform global retrieval over the same source-domain memory and reliably align samples to this universal anomaly pattern via their relative distances, yielding stronger cross-class and cross-domain generalization and more discriminative similarity scores. Fig. 2 in Sec. 3.1.1 further demonstrates cross-domain positive anomaly similarities and contrasts the separability before and after fine-tuning.

Table 4: Comparison with recent SOTA methods. We report I-AUROC, P-AUROC, inference time per image (ms), and training cost measured on a single RTX 3090 (trainable parameters, per-epoch time, model size, and peak GPU memory). Red = best, blue = second best.

| Method | Accuracy & inference | | | Training cost (RTX 3090) | | | |
|---|---|---|---|---|---|---|---|
| | I-AUROC | P-AUROC | Inf. (ms/img) | Params | Time / epo (s) | Model-size (MB) | Mem (GB) |
| WinCLIP | 75.1 | 73.0 | 840.3 | – | – | – | – |
| MRAD-TF (ours) | 81.0 | 85.5 | 198.3 | – | – | – | – |
| AdaCLIP | 87.8 | 85.6 | 226.2 | 10,665,472 | 753 | 41 | 12.0 |
| AnomalyCLIP | 90.1 | 91.6 | 177.6 | 5,555,200 | 341 | 22 | 7.9 |
| FAPrompt | 91.3 | 90.7 | 233.1 | 9,612,256 | 449 | 39 | 11.0 |
| MRAD-FT (ours) | 92.0 | 91.9 | 198.8 | 2,755,584 | 334 | 10 | 7.8 |
| MRAD-CLIP (ours) | 92.7 | 92.7 | 203.0 | 9,491,968 | 353 | 54 | 9.4 |

## C.3 MORE COMPARISON WITH STATE-OF-THE-ART METHODS

All statistics in Table 4 are obtained under a unified protocol. The number of trainable parameters is taken directly from the official implementations of each method. Performance metrics (I-AUROC / P-AUROC) are reported as the average results across all 16 datasets used in our benchmark. Inference time is measured on the MVTec-AD dataset, averaged per image, with the GPU kept in an idle state to ensure fair comparison. Training-time and memory statistics on the right are measured on a single RTX 3090, and the reported values correspond to per-epoch wall-clock time, checkpoint size on disk, and peak GPU memory usage during training.

From Table 4, MRAD-CLIP attains the highest overall detection accuracy, while MRAD-FT delivers the second-best performance with far fewer trainable parameters than other learnable baselines. In addition to accuracy, our models are economical to train: MRAD-FT uses a much smaller checkpoint and lower peak GPU memory, and converges within a single epoch; MRAD-CLIP also converges in only a few epochs while keeping training overhead moderate. By contrast, existing prompt-tuning methods such as AdaCLIP, AnomalyCLIP, and FAPrompt rely on larger parameter budgets, higher memory usage, and longer training schedules. Overall, these results indicate that the MRAD framework is both lightweight and training-friendly, achieving state-of-the-art performance under substantially reduced computational and resource costs.

## C.4 EXTENDED ABLATIONS

**Ablation on dynamic prompt biases.** In the main paper we ablated the source of additive bias for MRAD-CLIP's *dynamic prompts* while keeping all other components fixed. We observed that conditioning prompts on region-level context improves ZSAD. Motivated by these findings, we further provide a comprehensive comparison of three bias sources on all datasets in our benchmark: (i) *cross-patch* context, implemented by two learnable queries attending to patch tokens via a cross-attention mechanism, (ii) *anomaly prior* distilled from MRAD-FT, and (iii) *dual prior* that injects both normal and anomalous priors from MRAD-FT. The implementation strictly follows the main protocol with the same backbone, only the bias source is varied, while prompt length and all other hyperparameters are kept fixed. Tables 5 and 6 report the pixel-level (AUROC/PRO) and image-level (AUROC/AP) results of different bias sources across all datasets. It can be observed that our *dual prior* design in MRAD-CLIP consistently achieves the best overall performance, outperforming both the cross-patch and anomaly prior variants. It is also noteworthy that all three variants surpass AnomalyCLIP (Zhou et al., 2024), the representative ZSAD method with static prompts, highlighting the effectiveness of introducing learnable dynamic priors into the textual space.

**Ablation on mask threshold.** We conduct ablation experiments on the similarity-masking threshold $\rho$ in MRAD-FT by varying it as a hyperparameter, while keeping all other components fixed. For rigor, image-level evaluation is performed solely with $Y_{cls}$, whereas pixel-level evaluation directly uses $Y_{seg}$. As shown in Figure 11, moderate masking not only prevents overfitting caused by trivial self-matching in single-dataset settings, but also encourages the model to search for consistent abnormal cues across less similar features. Based on these results, we set $\rho_{seg} = 20\%$ for segmentation and $\rho_{cls} = 5\%$ for classification in all subsequent experiments.

Table 5: Pixel-level results (AUROC / PRO) for three bias sources across datasets.

| Dataset | Cross-patch | Anomaly prior | Dual prior |
|---|---|---|---|
| *Industrial* | | | |
| MVTec-AD | 92.5 / 85.7 | 92.8 / 85.2 | 93.0 / 86.8 |
| VisA | 95.3 / 88.6 | 95.6 / 88.1 | 95.9 / 88.0 |
| BTAD | 94.4 / 68.2 | 95.1 / 71.1 | 95.4 / 72.8 |
| MPDD | 97.8 / 90.3 | 97.7 / 88.8 | 97.9 / 90.6 |
| DTD-Synthetic | 97.5 / 90.7 | 98.4 / 90.3 | 98.1 / 89.8 |
| SDD | 90.2 / 72.0 | 87.3 / 67.8 | 93.0 / 72.0 |
| KSDD2 | 97.8 / 93.3 | 97.9 / 93.8 | 98.9 / 95.6 |
| DAGM | 95.4 / 88.2 | 96.6 / 90.1 | 97.4 / 90.3 |
| *Medical* | | | |
| ISIC | 90.7 / 82.5 | 91.7 / 83.9 | 91.3 / 83.4 |
| CVC-ColonDB | 84.6 / 72.0 | 84.7 / 74.0 | 84.7 / 73.9 |
| CVC-ClinicDB | 85.3 / 70.7 | 86.8 / 73.0 | 87.3 / 73.9 |
| Kvasir | 83.0 / 50.0 | 83.7 / 51.8 | 84.3 / 52.7 |
| Endo | 87.5 / 68.9 | 88.2 / 71.1 | 88.3 / 71.6 |
| **Average** | **91.7 / 78.6** | **92.0 / 79.2** | **92.7 / 80.1** |

Table 6: Image-level results (AUROC / AP) for three bias sources across datasets.

| Dataset | Cross-patch | Anomaly prior | Dual prior |
|---|---|---|---|
| *Industrial* | | | |
| MVTec-AD | 93.0 / 96.9 | 93.6 / 97.3 | 94.0 / 97.4 |
| VisA | 85.2 / 88.4 | 85.6 / 88.2 | 85.7 / 88.3 |
| BTAD | 93.1 / 97.8 | 92.0 / 94.7 | 92.8 / 94.2 |
| MPDD | 83.4 / 86.2 | 81.7 / 83.6 | 81.8 / 83.4 |
| DTD-Synthetic | 94.2 / 97.7 | 95.9 / 98.4 | 96.0 / 98.4 |
| SDD | 82.6 / 73.1 | 81.8 / 69.6 | 83.9 / 76.2 |
| KSDD2 | 92.4 / 86.0 | 92.9 / 85.8 | 95.1 / 88.9 |
| DAGM | 97.8 / 98.6 | 97.9 / 98.1 | 98.4 / 98.6 |
| *Medical* | | | |
| HeadCT | 97.4 / 97.6 | 96.5 / 96.7 | 97.1 / 97.6 |
| BrainMRI | 96.9 / 97.4 | 96.7 / 97.2 | 97.0 / 97.4 |
| Br35H | 97.7 / 97.1 | 97.4 / 97.1 | 97.9 / 97.6 |
| **Average** | **92.1 / 92.4** | **92.0 / 91.5** | **92.7 / 92.5** |

**Ablation on the two-level memory bank.** To understand the role of each branch, we perform ablations by selectively removing one level of memory at a time. First, when we discard the image-level memory and retain only the pixel-level memory, the image-level anomaly score is obtained by aggregating pixel-wise scores via top-$K$ pooling. Compared with the full dual-level memory, this variant consistently yields lower I-AUROC on most datasets (Fig. 12), sometimes with a noticeable margin, indicating that the image-level memory provides complementary and more stable global evidence on top of pixel-level anomalies. Conversely, when we remove the pixel-level memory and only keep the image-level branch, the model loses meaningful localization ability and the image-level performance also degrades, since it no longer benefits from the compensating top-$K$ aggregation from the segmentation branch. Together, these ablations show that strong and robust image-level detection relies on the complementary contribution of both image-level and pixel-level memories.

## C.5 EFFECT OF CHANGING THE AUXILIARY DATASET

Beyond the default setting where VisA is used as the auxiliary dataset, we also study how MRAD behaves when the memory bank is built from other sources. For each candidate auxiliary dataset in

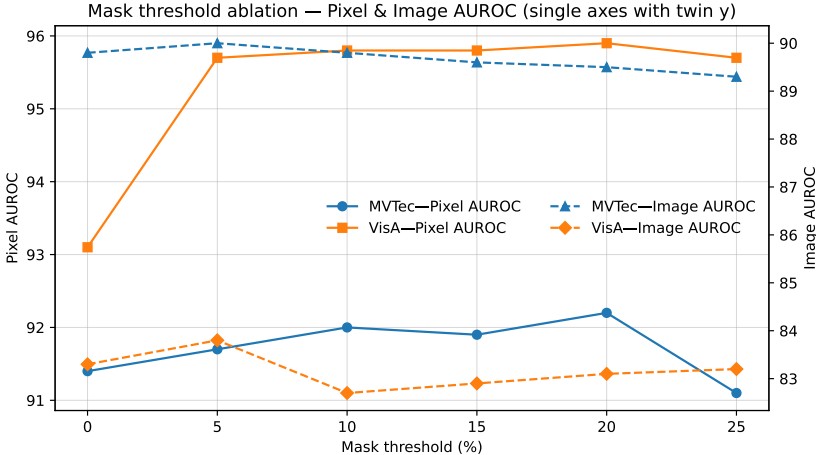

Figure 11: Metrics about the ablation on mask threshold.

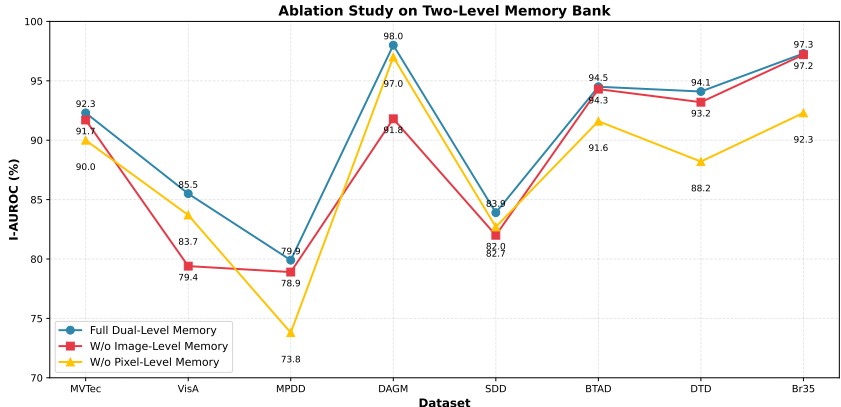

Figure 12: **Ablation on two-level memory bank.**

Table 7, we train MRAD-FT and MRAD-CLIP using this dataset alone to construct the memory and then evaluate zero-shot anomaly detection on MVTec-AD.

The key factor here is not whether the auxiliary dataset is "larger" or "stronger" in an absolute sense, but whether it allows MRAD to learn a cross-class anomaly pattern that can be reused at test time. MRAD's training objective requires each query to retrieve and compare across prototypes from multiple classes, so the auxiliary source must contain sufficiently diverse normal and anomalous samples for a class-agnostic anomaly pattern to emerge in the source-domain feature space. This explains why industrial datasets with reasonable category or defect-shape diversity (e.g., VisA, BTAD, MPDD, DTD-Synthetic) all support relatively stable zero-shot performance in Table 7: their prototypes cover multiple modes of industrial appearance, allowing the model to repeatedly "use anomalies from class A to retrieve against classes B/C" during training and thus form a transferable anomaly pattern.

Under these auxiliary sources, pixel-level performance remains consistently strong and generally surpasses AnomalyCLIP, whereas image-level performance exhibits larger variation. This variation mainly arises from the class token being more sensitive to global image statistics: for example, DTD-Synthetic is a texture-style dataset with 12 nominal classes but highly similar backgrounds and appearances, so its class tokens provide limited semantic diversity and the resulting image-level memory tends to encode repetitive texture patterns rather than discriminative semantics, thereby weakening image-level classification.

Table 7: Effect of changing the auxiliary dataset used to build the memory bank. Each auxiliary dataset is used alone for training, and we report zero-shot performance on MVTec-AD (P-AUROC / I-AUROC). For reference, AnomalyCLIP trained on VisA achieves 91.1 / 91.5.

| Domain | Aux. dataset | Classes | Normal | Anomaly | MRAD-FT | MRAD-CLIP |
|---|---|---|---|---|---|---|
| Industrial | **VisA** | **12** | **962** | **1200** | **92.2 / 92.3** | **93.0 / 94.0** |
| | BTAD | 3 | 451 | 290 | 92.0 / 92.5 | 91.0 / 92.2 |
| | MPDD | 6 | 176 | 282 | 91.1 / 87.3 | 91.2 / 89.2 |
| | DTD-Synthetic | 12 | 357 | 947 | 91.3 / 88.7 | 91.5 / 89.8 |
| | SDD | 1 | 181 | 74 | 88.1 / 82.5 | 88.9 / 87.8 |
| | KSDD2 | 1 | 894 | 110 | 90.2 / 86.3 | 92.1 / 92.8 |
| Medical | CVC-ClinicDB | 1 | 0 | 612 | 87.5 / 75.0 | 88.1 / 79.7 |
| | Kvasir | 1 | 0 | 1000 | 86.0 / 70.6 | 84.6 / 73.1 |

In contrast, when the auxiliary dataset collapses to essentially a single industrial category (e.g., SDD, KSDD2), the model has almost no opportunity to practice genuine cross-class retrieval during training. The anomaly prototypes then only capture one specific defect morphology and fail to induce a transferable anomaly pattern; consequently, MRAD-FT shows more noticeable degradation on MVTec-AD. The situation becomes even more extreme when the auxiliary source is both single-category and strongly out-of-domain, as with medical datasets such as CVC-ClinicDB and Kvasir: their prototype distributions differ substantially from industrial imagery, so the memory bank lacks both cross-class contrast and appearance statistics aligned with the target domain. In this regime, cross-domain retrieval becomes intrinsically difficult and the performance drop is the most pronounced.

Overall, Table 7 indicates that MRAD is not tied to VisA or any particular auxiliary dataset. As long as the auxiliary source lies within a related domain and provides sufficient category and appearance diversity, MRAD-FT and MRAD-CLIP can learn a class-agnostic anomaly pattern in the source domain and maintain relatively stable performance under cross-dataset evaluation. Significant degradation mainly occurs when the auxiliary dataset offers very limited semantic coverage or exhibits a large domain shift relative to the target, but even in these extreme cases the model does not collapse, which is consistent with our design intuition that MRAD relies on learning a transferable anomaly pattern from a sufficiently diverse prototype space rather than on a specific dataset choice.

## C.6 ADDITIONAL DATASET EXPERIMENTS

Table 8: Image-level performance on the Liver CT ZSAD benchmark (I-AUROC / I-AP).

| Method | I-AUROC | I-AP |
|---|---|---|
| WinCLIP | 63.7 | 52.7 |
| MRAD-TF | 66.3 | **62.1** |
| AdaCLIP | 61.6 | 52.5 |
| AnomalyCLIP | 63.6 | 60.7 |
| FAPrompt | 65.0 | 60.1 |
| MRAD-FT | 67.0 | 60.2 |
| MRAD-CLIP | **67.5** | 60.9 |

To further examine the applicability of MRAD in medical imaging scenarios, we additionally include a Liver CT dataset from BMAD Bao et al. (2024), which contains 1,493 CT slices with image-level anomaly annotations. Following the same ZSAD protocol as in the main paper, we evaluate several representative baselines together with our MRAD variants. Table 8 reports image-level AUROC and AP.

Across all methods, MRAD-FT and MRAD-CLIP achieve the strongest AUROC and competitive AP, indicating that the proposed retrieval-based memory framework can generalize beyond industrial imaging and remain effective under medical CT imagery with different appearance statistics.

## C.7 STABILITY AND ROBUSTNESS ANALYSIS

Table 9: Stability across random seeds on MVTec-AD. We report mean P-AUROC and I-AUROC over multiple runs and the maximum deviation from the mean (in percentage points).

| Method | P-AUROC (mean) | I-AUROC (mean) | Max deviation |
|---|---|---|---|
| MRAD-FT | 92.2 | 91.9 | $\leq 0.5$ |
| MRAD-CLIP | 92.7 | 93.6 | $\leq 0.4$ |

Table 10: Robustness to test-time perturbations on MVTec-AD. We report image-level and pixel-level performance on the original test set and on a perturbed version with additive noise and brightness adjustment.

| | Original Test Set | | Noisy + Brightness | |
|---|---|---|---|---|
| Method | P-AUROC | I-AUROC | P-AUROC | I-AUROC |
| MRAD-FT | 92.2 | 92.3 | 91.6 | 91.1 |
| MRAD-CLIP | 93.0 | 94.0 | 92.6 | 92.8 |

We further examine the stability of MRAD with respect to random initialization and its robustness under simple distribution shifts.

**Variance across random seeds.** To quantify the effect of randomness, we run MRAD-FT and MRAD-CLIP on MVTec-AD with multiple random seeds and report the mean performance across runs. As shown in Table 9, both P-AUROC and I-AUROC exhibit very small fluctuations (within $\pm 0.5$ percentage points), and the means closely match the single-run numbers reported in the main paper.

**Robustness to simple distribution shifts.** We then evaluate robustness to common test-time perturbations by applying a random brightness increase or decrease of $15\%$ and additive Gaussian noise with a standard deviation of $5$ to the MVTec-AD test images, and re-evaluating the models on the perturbed set. Table 10 summarizes the results. Both MRAD-FT and MRAD-CLIP show only mild degradation, with drops in P-AUROC and I-AUROC below 1 percentage point. MRAD-CLIP is slightly more robust overall, and its performance under these perturbations remains higher than that of prior prompt-based baselines on the clean MVTec-AD set. These findings suggest that MRAD maintains reasonable robustness to typical illumination and low-level noise shifts and does not collapse under moderate degradation of image quality.

## C.8 INSTANTIATING ANOMALY PROTOTYPES FROM SYNTHETIC PATCHES

Table 11: Effect of replacing real anomalies with synthetic patches when constructing anomaly prototypes.

| Method | Anomaly source | P-AUROC | I-AUROC |
|---|---|---|---|
| MRAD-FT | Real anomalies | 92.2 | 92.3 |
| | Synthetic patches | 89.2 | 79.7 |
| MRAD-CLIP | Real anomalies | 93.0 | 94.0 |
| | Synthetic patches | 83.1 | 76.1 |

A natural question is whether MRAD-TF/FT can be instantiated without real anomalies, using only normal images (without masks) with synthetic anomaly patches. This setting probes how far MRAD can reduce its dependence on real defect samples.

Concretely, we use VisA as the auxiliary dataset to construct normal prototypes, and follow a DRAEM-style Zavrtanik et al. (2021) local perturbation scheme (Cut&Paste) to synthesize anomalous regions on normal images. These synthetic regions are then used to build the anomaly memory, while all other training and evaluation settings are kept unchanged. We finally evaluate zero-shot anomaly detection on MVTec-AD. Table 11 compares the performance when anomaly prototypes are constructed from real anomalies (the default setting in the main paper) versus from synthetic patches.

As shown in Table 11, replacing real anomalies with synthetic patches leads to a noticeable degradation, especially for MRAD-CLIP at the image level. This is consistent with our intuition: the retrieval mechanism in MRAD relies on the anomaly prototypes in the memory bank to cover real defect patterns, while synthetic anomalies mainly mimic local texture perturbations and lack the semantic diversity of real industrial defects, making cross-category retrieval less reliable.

Overall, operating in the "normal data + synthetic anomalies" regime leads to a clear degradation in performance under our current configuration, yet the setting remains informative from a weak-supervision standpoint. Even when anomaly prototypes are derived solely from synthetic patches, MRAD-FT still yields segmentation performance at a practically usable level, indicating that a memory-based retrieval framework can, in principle, function in the absence of real anomalies and constitutes a promising basis for extending ZSAD to purely synthetic-supervision scenarios.

## C.9 Fine-grained ZSAD performance

In this section, we report fine-grained subset-level ZSAD performance on the representative MVTec-AD and VisA datasets, with per-category results summarized in Tables 13–20. Compared to the aggregated metrics in the main text, these detailed breakdowns offer a clearer view of category-level performance, highlighting the robustness of MRAD-CLIP and its variants across diverse anomaly types.

## C.10 Comparison with additional baseline

To further evaluate the contribution of the proposed memory-retrieval design, an additional baseline was constructed by equipping AnomalyCLIP with a simple "vanilla" memory bank. This variant follows the general idea of reference-based matching used in prior few-shot anomaly detection methods, but adapted to the zero-shot setting. The goal is to isolate the effect of incorporating a memory mechanism while preserving AnomalyCLIP's original prompt–patch inference pipeline.

The implementation keeps AnomalyCLIP's original zero-shot prediction (based on prompt–patch cosine similarity). In addition, a patch-level memory bank is built using all samples from the auxiliary dataset (VisA), following the setting used in our retrieval-based methods. For each query patch, a nearest-neighbor discrepancy score is computed against this reference patch set. The final anomaly score is obtained by combining this discrepancy with AnomalyCLIP's zero-shot output. This design represents the simplest and most direct form of a vanilla memory mechanism integrated into AnomalyCLIP, without modifying its architecture or training strategy.

Table 12 reports the pixel-level AUROC on seven datasets. We find that adding this vanilla memory bank to AnomalyCLIP does not bring meaningful improvement. On some datasets (e.g., BTAD, ClinicDB), the performance even drops. We believe this mainly happens because nearest-neighbor patch retrieval works in FSAD, where reference and query images usually share the same object or background, but it does not transfer to ZSAD, where categories differ and cross-dataset patch matching becomes unreliable.

In contrast, both MRAD-FT and MRAD-CLIP achieve consistently stronger performance. This advantage stems from the fact that MRAD explicitly stores discriminative feature–label prototypes and supports cross-category, cross-domain retrieval, which is crucial for generalizable zero-shot anomaly detection. These results further validate the effectiveness of the proposed memory-retrieval design as a key component of the MRAD framework.

Table 12: Pixel-level AUROC comparison between AnomalyCLIP, AnomalyCLIP with a vanilla memory bank, and the MRAD variants.

| Dataset | AnomalyCLIP | AnomalyCLIP + VanillaMB | MRAD-FT | MRAD-CLIP |
|---------|-------------|-------------------------|---------|-----------|
| MVTec | 91.1 | 91.0 | 92.2 | 93.0 |
| BTAD | 93.3 | 89.2 | 94.7 | 95.4 |
| MPDD | 96.2 | 96.5 | 97.4 | 97.9 |
| SDD | 90.1 | 90.3 | 91.0 | 93.0 |
| KSDD2 | 97.9 | 98.2 | 98.8 | 98.9 |
| Kvasir | 81.9 | 82.0 | 83.9 | 84.3 |
| ClinicDB | 85.9 | 85.3 | 85.9 | 87.3 |

Table 13: Fine-grained pixel-level AUROC (%) on MVTec-AD. Best per row in **bold**.

| Category | WinCLIP | MRAD-TF | AdaCLIP | AnomalyCLIP | FAPrompt | MRAD-FT | MRAD-CLIP |
|----------|---------|---------|---------|-------------|----------|---------|-----------|
| bottle | 89.5 | 81.9 | 83.8 | 90.4 | 90.3 | 90.6 | **92.2** |
| cable | 77.0 | 70.1 | **85.6** | 78.9 | 79.5 | 78.7 | 78.4 |
| capsule | 86.9 | 90.1 | 86.2 | 95.8 | 95.2 | 97.0 | **97.8** |
| carpet | 95.4 | 97.0 | 94.8 | 98.8 | 99.0 | **99.5** | 99.4 |
| grid | 82.2 | 93.0 | 90.6 | 97.3 | 96.9 | 98.1 | **98.4** |
| hazelnut | 94.3 | 90.1 | **98.7** | 97.2 | 97.5 | 97.1 | 97.5 |
| leather | 96.7 | 99.0 | 97.8 | 98.6 | 98.5 | 99.0 | **99.1** |
| metal_nut | 61.0 | 75.6 | 55.4 | 74.6 | 71.4 | 74.3 | **79.4** |
| pill | 80.0 | 81.0 | 77.5 | 91.8 | 90.5 | **92.8** | 91.3 |
| screw | 89.6 | 91.3 | **99.2** | 97.5 | 97.4 | 97.5 | 97.6 |
| tile | 77.6 | 93.7 | 83.9 | 94.7 | 95.7 | 96.8 | **97.0** |
| toothbrush | 86.9 | 87.7 | 93.4 | 91.9 | 89.7 | **96.5** | 96.1 |
| transistor | 74.7 | 67.0 | 71.4 | 70.8 | 69.8 | 71.2 | **75.0** |
| wood | 93.4 | 96.4 | 91.2 | 96.4 | 96.4 | **98.1** | 98.0 |
| zipper | 91.6 | 86.9 | 91.8 | 91.2 | 91.8 | 95.2 | **96.7** |
| mean | 85.1 | 86.7 | 86.8 | 91.1 | 90.6 | 92.2 | **93.0** |

## D  VISUALIZATION

In this section, we report the anomaly map visualizations obtained from MRAD-CLIP across all datasets used in our experiments (Figure 13–Figure 37). These examples cover representative categories from each dataset, providing a more detailed view of the model's anomaly segmentation capability. The results highlight the robustness of our method in capturing fine-grained defects and domain-specific anomalies across both industrial and medical scenarios.

## E  USE OF LARGE LANGUAGE MODELS (LLMs)

In preparing this manuscript, we made limited use of large language models (LLMs), specifically for translation, grammar correction, and minor language polishing. The LLMs were not involved in research ideation, experimental design, data analysis, or substantive writing of the scientific content.

Table 14: Fine-grained pixel-level PRO (%) on MVTec-AD. Best per row in **bold**.

| Category | WinCLIP | MRAD-TF | AdaCLIP | AnomalyCLIP | FAPrompt | MRAD-FT | MRAD-CLIP |
|---|---|---|---|---|---|---|---|
| bottle | 76.4 | 62.5 | 26.9 | 80.8 | 81.0 | 82.0 | **84.7** |
| cable | 42.9 | 24.7 | 15.2 | 64.0 | 68.2 | 65.3 | **69.8** |
| capsule | 62.1 | 57.8 | 65.7 | 87.6 | 83.9 | 92.4 | **94.7** |
| carpet | 84.1 | 89.9 | 19.6 | 90.0 | 94.1 | **98.0** | 96.5 |
| grid | 57.0 | 78.5 | 46.2 | 75.4 | 81.6 | **92.7** | 84.8 |
| hazelnut | 81.6 | 67.5 | 42.3 | 92.5 | **93.3** | 92.2 | 92.9 |
| leather | 91.1 | 97.3 | 55.9 | 92.2 | 95.7 | 97.1 | **97.8** |
| metal_nut | 31.8 | 38.9 | 20.7 | 71.1 | 70.9 | 68.0 | **75.7** |
| pill | 65.0 | 48.3 | 37.0 | 88.1 | 87.6 | 91.2 | **92.6** |
| screw | 68.5 | 68.9 | 75.3 | 88.0 | 89.7 | 89.4 | **90.4** |
| tile | 51.2 | 88.5 | 7.7 | 87.4 | 89.3 | **93.1** | 91.3 |
| toothbrush | 67.7 | 42.0 | 25.6 | 88.5 | 87.3 | 87.0 | **89.5** |
| transistor | 43.4 | 43.9 | 6.7 | 58.2 | **59.0** | 55.1 | 58.7 |
| wood | 74.1 | 91.1 | 58.3 | 91.5 | 92.3 | **95.0** | 94.3 |
| zipper | 71.7 | 52.6 | 3.4 | 65.4 | 75.1 | 82.4 | **88.2** |
| mean | 64.6 | 63.5 | 33.8 | 81.4 | 83.3 | 85.4 | **86.8** |

Table 15: Fine-grained image-level AUROC (%) on MVTec-AD. Best per row in **bold**.

| Category | WinCLIP | MRAD-TF | AdaCLIP | AnomalyCLIP | FAPrompt | MRAD-FT | MRAD-CLIP |
|---|---|---|---|---|---|---|---|
| bottle | **99.2** | 81.5 | 95.6 | 88.7 | 89.8 | 91.9 | 92.7 |
| cable | 86.5 | 68.2 | 79.0 | 70.3 | 74.7 | 81.8 | **92.0** |
| capsule | 72.9 | 78.6 | 89.3 | 89.5 | 92.4 | 92.3 | **96.4** |
| carpet | **100.0** | 96.3 | **100.0** | 99.9 | **100.0** | **100.0** | **100.0** |
| grid | 98.8 | 92.1 | **99.2** | 97.8 | 97.9 | 99.1 | 98.9 |
| hazelnut | 93.9 | 65.6 | 95.5 | **97.2** | 96.5 | 93.9 | 93.0 |
| leather | **100.0** | **100.0** | **100.0** | 99.8 | 99.9 | **100.0** | **100.0** |
| metal_nut | **97.1** | 33.9 | 79.9 | 92.4 | 89.7 | 71.3 | 78.8 |
| pill | 79.1 | 69.1 | **92.6** | 81.1 | 89.6 | 86.4 | 85.8 |
| screw | 83.3 | 64.5 | 83.9 | 82.1 | 85.0 | **87.5** | 86.7 |
| tile | **100.0** | 99.6 | 99.7 | **100.0** | 99.7 | 99.7 | 99.7 |
| toothbrush | 87.5 | 79.4 | 95.2 | 85.3 | 85.6 | 95.6 | **97.2** |
| transistor | 88.0 | 71.2 | 82.0 | **93.9** | 81.7 | 86.8 | 89.1 |
| wood | **99.4** | 99.3 | 98.5 | 96.9 | 98.0 | 98.9 | 99.0 |
| zipper | 91.5 | 85.6 | 89.4 | 98.4 | 98.4 | 99.6 | **99.9** |
| mean | 91.8 | 79.0 | 92.0 | 91.5 | 91.9 | 92.3 | **94.0** |

Table 16: Fine-grained image-level AP (%) on MVTec-AD. Best per row in **bold**.

| Category | WinCLIP | MRAD-TF | AdaCLIP | AnomalyCLIP | FAPrompt | MRAD-FT | MRAD-CLIP |
|---|---|---|---|---|---|---|---|
| bottle | 98.3 | 94.1 | **98.6** | 96.8 | 96.7 | 97.7 | 97.9 |
| cable | 86.2 | 80.4 | 87.3 | 81.7 | 82.9 | 89.5 | **95.4** |
| capsule | 93.4 | 94.0 | 97.8 | 97.8 | 98.4 | 98.4 | **99.2** |
| carpet | 99.9 | 98.9 | **100.0** | 99.9 | **100.0** | **100.0** | **100.0** |
| grid | **99.8** | 97.3 | 99.7 | 99.3 | 99.3 | 99.7 | 99.6 |
| hazelnut | 96.3 | 77.5 | 97.5 | **98.5** | 98.1 | 97.0 | 96.6 |
| leather | **100.0** | **100.0** | **100.0** | 99.9 | **100.0** | **100.0** | **100.0** |
| metal_nut | 97.9 | 76.0 | 95.6 | **98.1** | 97.5 | 93.3 | 95.2 |
| pill | 96.5 | 89.5 | **98.6** | 95.3 | 97.9 | 97.1 | 97.0 |
| screw | 88.4 | 84.5 | 93.0 | 92.9 | 93.6 | **95.4** | 94.5 |
| tile | 99.9 | 99.8 | 99.9 | **100.0** | 99.9 | 99.9 | 99.9 |
| toothbrush | 96.7 | 91.8 | 97.9 | 93.9 | 93.8 | 98.3 | **98.9** |
| transistor | 74.9 | 70.6 | 83.8 | **92.1** | 78.9 | 83.6 | 86.4 |
| wood | 98.8 | **99.8** | 99.5 | 99.2 | 99.4 | 99.7 | 99.7 |
| zipper | 98.9 | 95.8 | 97.1 | 99.5 | 99.5 | 99.9 | **100.0** |
| mean | 95.1 | 90.0 | 96.4 | 96.2 | 95.7 | 96.6 | **97.4** |

Table 17: Fine-grained pixel-level AUROC (%) on VisA. Best per row in **bold**.

| Category | WinCLIP | MRAD-TF | AdaCLIP | AnomalyCLIP | FAPrompt | MRAD-FT | MRAD-CLIP |
|---|---|---|---|---|---|---|---|
| candle | 88.9 | 95.0 | 98.6 | 98.8 | **98.9** | **98.9** | **98.9** |
| capsules | 81.6 | 84.7 | 96.1 | 94.9 | 96.3 | **96.5** | 95.4 |
| cashew | 84.7 | 92.0 | 97.2 | 93.7 | 95.2 | 95.1 | **97.9** |
| chewinggum | 93.3 | 98.1 | 99.2 | 99.2 | 99.3 | **99.5** | **99.5** |
| fryum | 88.5 | 92.5 | 93.6 | 94.6 | 94.4 | **95.2** | 94.3 |
| macaroni1 | 70.9 | 91.9 | 98.8 | 98.3 | 98.2 | **99.0** | 98.8 |
| macaroni2 | 59.3 | 93.7 | **98.2** | 97.6 | 96.8 | 97.7 | 97.6 |
| pcb1 | 61.2 | 81.9 | 90.7 | 94.0 | **96.0** | 92.5 | 92.3 |
| pcb2 | 71.6 | 86.8 | 91.3 | 92.4 | 92.7 | **92.9** | 91.4 |
| pcb3 | 85.3 | 86.9 | 87.7 | 88.3 | 88.2 | **88.9** | 88.7 |
| pcb4 | 94.4 | 92.8 | 94.6 | 95.7 | 97.1 | **96.8** | **96.8** |
| pipe_fryum | 75.4 | 95.6 | 95.7 | **98.2** | 98.1 | 97.4 | **98.2** |
| mean | 79.6 | 91.0 | 95.1 | 95.5 | **95.9** | **95.9** | **95.9** |

Table 18: Fine-grained pixel-level PRO (%) on VisA. Best per row in **bold**.

| Category | WinCLIP | MRAD-TF | AdaCLIP | AnomalyCLIP | FAPrompt | MRAD-FT | MRAD-CLIP |
|---|---|---|---|---|---|---|---|
| candle | 83.5 | 91.2 | 71.6 | **96.5** | 95.8 | 95.9 | 94.6 |
| capsules | 35.3 | 47.3 | 80.3 | 78.9 | 84.9 | **85.5** | 79.4 |
| cashew | 76.4 | 72.5 | 45.6 | 91.9 | 90.0 | 92.1 | **93.8** |
| chewinggum | 70.4 | 82.4 | 53.9 | 90.9 | 90.1 | **93.9** | 91.8 |
| fryum | 77.4 | 69.1 | 55.6 | 86.8 | 87.1 | **89.1** | 88.5 |
| macaroni1 | 34.3 | 73.9 | 86.6 | 89.7 | 89.9 | 93.0 | **93.3** |
| macaroni2 | 21.4 | 77.0 | 84.8 | 83.9 | 80.3 | **85.1** | **85.1** |
| pcb1 | 26.3 | 43.8 | 52.3 | 80.7 | 87.3 | **88.1** | 84.3 |
| pcb2 | 37.2 | 63.0 | 77.5 | 78.2 | 77.8 | **80.4** | 76.0 |
| pcb3 | 56.1 | 67.8 | 76.2 | 76.8 | 77.8 | 79.3 | **82.1** |
| pcb4 | 80.4 | 80.6 | 84.3 | 89.4 | **91.7** | **91.7** | 90.3 |
| pipe_fryum | 82.3 | 83.3 | 86.3 | 96.1 | **97.2** | 95.2 | 96.4 |
| mean | 56.8 | 71.0 | 71.3 | 87.0 | 87.5 | **89.1** | 88.0 |

Table 19: Fine-grained image-level AUROC (%) on VisA. Best per row in **bold**.

| Category | WinCLIP | MRAD-TF | AdaCLIP | AnomalyCLIP | FAPrompt | MRAD-FT | MRAD-CLIP |
|---|---|---|---|---|---|---|---|
| candle | 95.4 | 73.9 | **95.9** | 80.9 | 87.2 | 82.0 | 82.8 |
| capsules | 85.0 | 73.4 | 81.1 | 82.7 | 91.6 | **93.0** | 91.5 |
| cashew | **92.1** | 79.7 | 89.6 | 76.0 | 90.5 | 89.3 | 90.6 |
| chewinggum | 96.5 | 98.1 | **98.5** | 97.2 | 97.6 | 97.7 | 97.6 |
| fryum | 80.3 | 86.5 | 89.5 | 92.7 | **96.5** | 93.2 | 93.5 |
| macaroni1 | 76.2 | 70.1 | 86.3 | **86.7** | 83.1 | 85.6 | 85.6 |
| macaroni2 | 63.7 | 62.2 | 56.7 | 72.2 | 71.4 | **79.4** | 78.4 |
| pcb1 | 73.6 | 55.1 | 74.0 | **85.2** | 68.2 | 83.3 | 83.4 |
| pcb2 | 51.2 | 62.9 | **71.1** | 62.0 | 66.4 | 66.3 | 64.8 |
| pcb3 | 73.4 | 65.6 | **75.2** | 61.7 | 68.6 | 64.7 | 68.9 |
| pcb4 | 79.6 | 95.1 | 89.6 | 93.9 | 95.4 | **97.1** | **97.1** |
| pipe_fryum | 69.7 | 76.9 | 88.8 | 92.3 | **97.4** | 94.1 | 94.3 |
| mean | 78.1 | 75.0 | 83.0 | 82.1 | 84.5 | 85.5 | **85.7** |

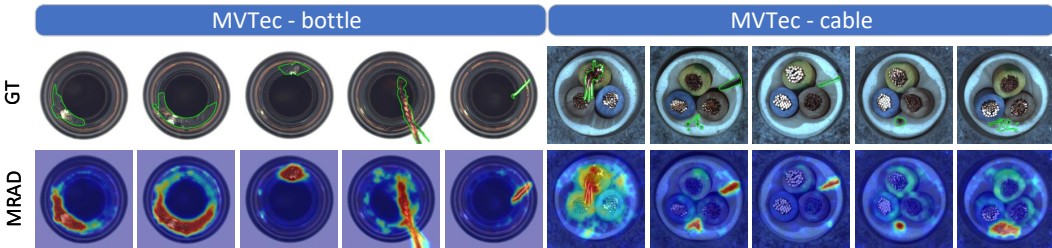

Figure 13: Anomaly maps of MRAD-CLIP across various categories. The first row shows the ground-truth annotations, and the second row shows the predicted anomaly maps.

Table 20: Fine-grained image-level AP (%) on VisA. Best per row in **bold**.

| Category | WinCLIP | MRAD-TF | AdaCLIP | AnomalyCLIP | FAPrompt | MRAD-FT | MRAD-CLIP |
|---|---|---|---|---|---|---|---|
| candle | 95.6 | 76.0 | **96.4** | 82.6 | 89.7 | 84.4 | 86.2 |
| capsules | 80.9 | 83.5 | 86.7 | 89.4 | **96.2** | 95.8 | 95.0 |
| cashew | 95.2 | 90.6 | 95.4 | 89.3 | 95.9 | 95.4 | **96.1** |
| chewinggum | 98.8 | 99.2 | **99.4** | 98.8 | 99.1 | 99.1 | 99.1 |
| fryum | 92.5 | 93.6 | 95.1 | 96.6 | **98.4** | 96.7 | 97.0 |
| macaroni1 | 64.5 | 68.4 | 85.0 | 85.5 | 82.5 | 84.7 | **85.7** |
| macaroni2 | 65.2 | 60.2 | 54.3 | 70.8 | 68.5 | **79.0** | 78.0 |
| pcb1 | 74.6 | 58.5 | 73.5 | **86.7** | 72.5 | 84.8 | 84.9 |
| pcb2 | 44.2 | 65.1 | **71.6** | 64.4 | 68.2 | 68.8 | 67.9 |
| pcb3 | 66.2 | 72.9 | **77.9** | 69.4 | 76.5 | 71.1 | 74.9 |
| pcb4 | 70.1 | 95.8 | 89.8 | 94.3 | 95.6 | 96.7 | **96.8** |
| pipe_fryum | 82.1 | 85.7 | 93.9 | 96.3 | **98.6** | 97.0 | 97.1 |
| mean | 77.5 | 79.1 | 84.9 | 85.4 | 86.8 | 87.8 | **88.3** |

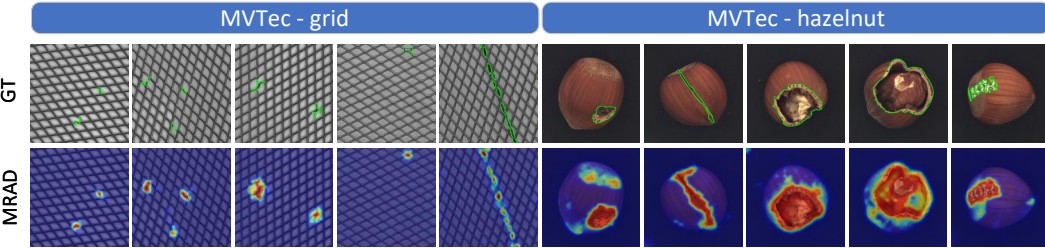

Figure 14: Anomaly maps of MRAD-CLIP across various categories. The first row shows the ground-truth annotations, and the second row shows the predicted anomaly maps.

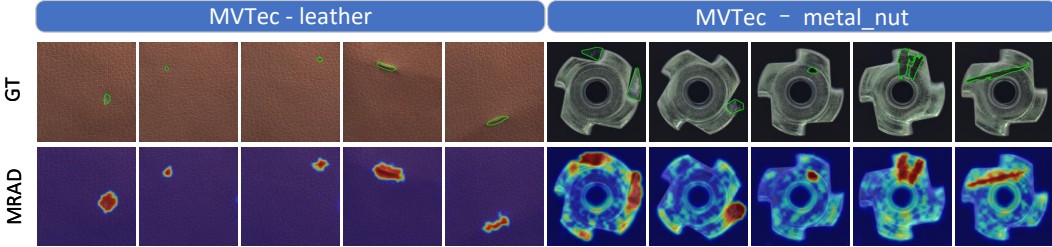

Figure 15: Anomaly maps of MRAD-CLIP across various categories. The first row shows the ground-truth annotations, and the second row shows the predicted anomaly maps.

Figure 16: Anomaly maps of MRAD-CLIP across various categories. The first row shows the ground-truth annotations, and the second row shows the predicted anomaly maps.

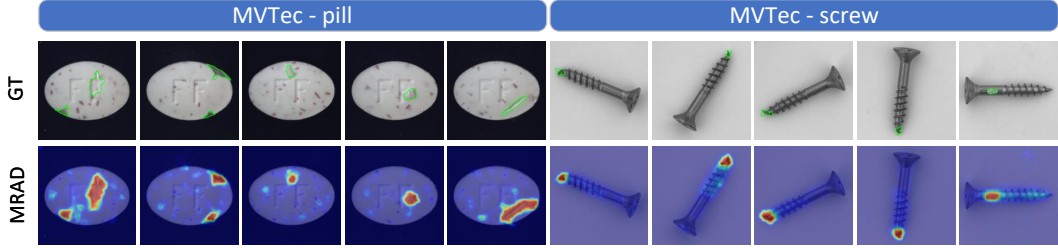

Figure 17: Anomaly maps of MRAD-CLIP across various categories. The first row shows the ground-truth annotations, and the second row shows the predicted anomaly maps.

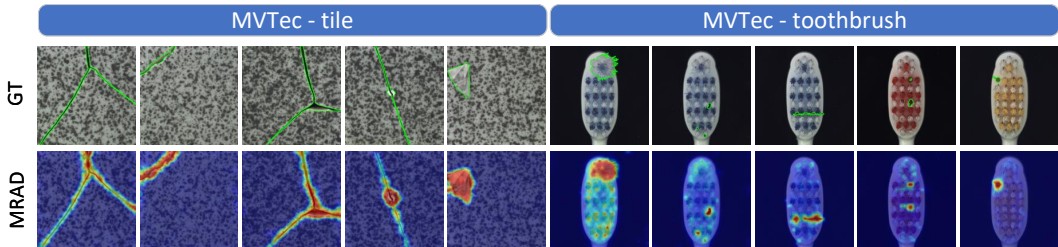

Figure 18: Anomaly maps of MRAD-CLIP across various categories. The first row shows the ground-truth annotations, and the second row shows the predicted anomaly maps.

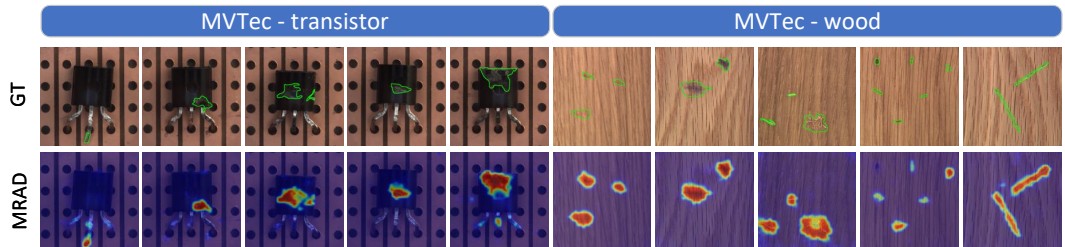

Figure 19: Anomaly maps of MRAD-CLIP across various categories. The first row shows the ground-truth annotations, and the second row shows the predicted anomaly maps.

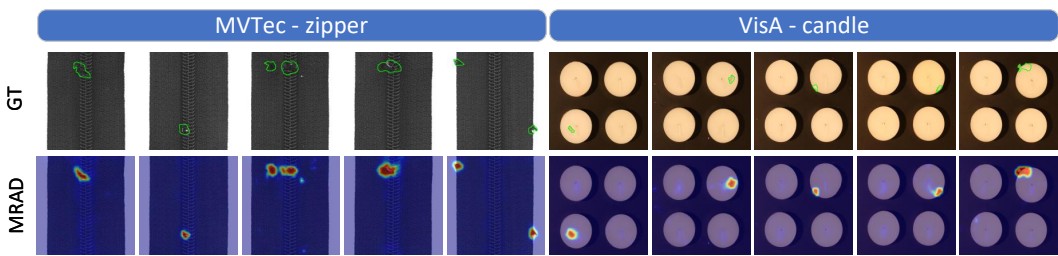

Figure 20: Anomaly maps of MRAD-CLIP across various categories. The first row shows the ground-truth annotations, and the second row shows the predicted anomaly maps.

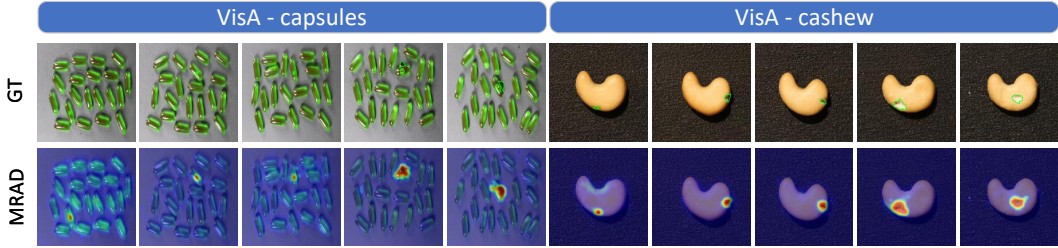

Figure 21: Anomaly maps of MRAD-CLIP across various categories. The first row shows the ground-truth annotations, and the second row shows the predicted anomaly maps.

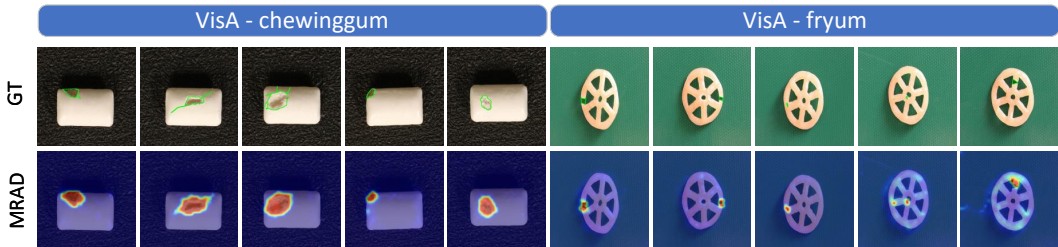

Figure 22: Anomaly maps of MRAD-CLIP across various categories. The first row shows the ground-truth annotations, and the second row shows the predicted anomaly maps.

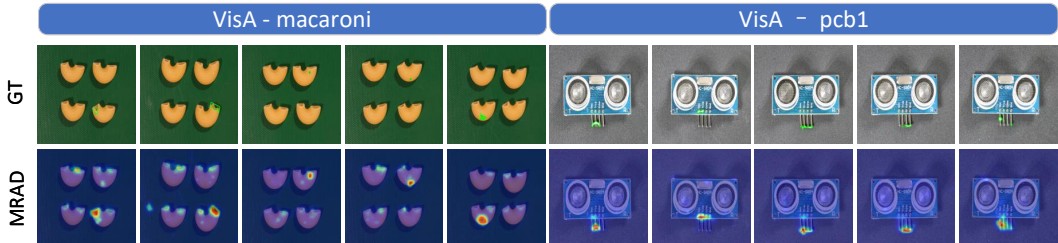

Figure 23: Anomaly maps of MRAD-CLIP across various categories. The first row shows the ground-truth annotations, and the second row shows the predicted anomaly maps.

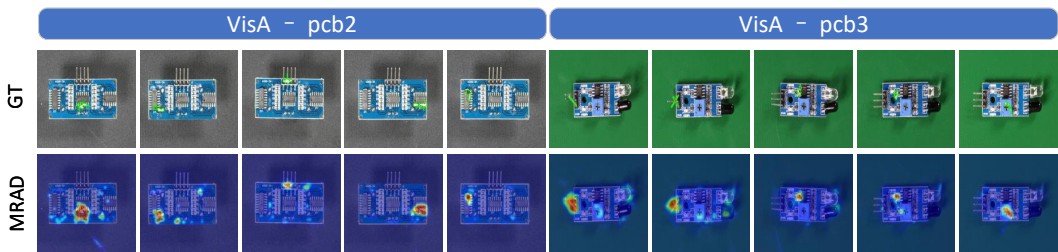

Figure 24: Anomaly maps of MRAD-CLIP across various categories. The first row shows the ground-truth annotations, and the second row shows the predicted anomaly maps.

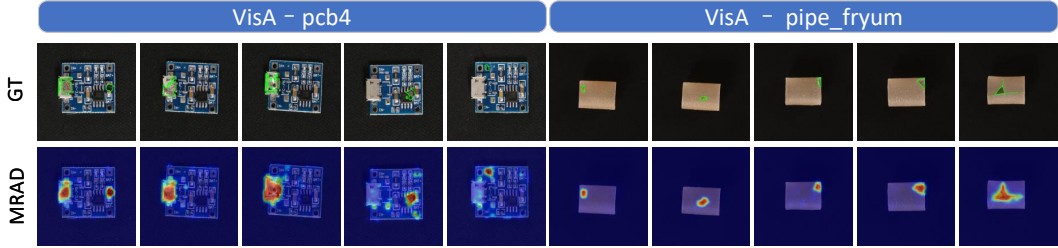

Figure 25: Anomaly maps of MRAD-CLIP across various categories. The first row shows the ground-truth annotations, and the second row shows the predicted anomaly maps.

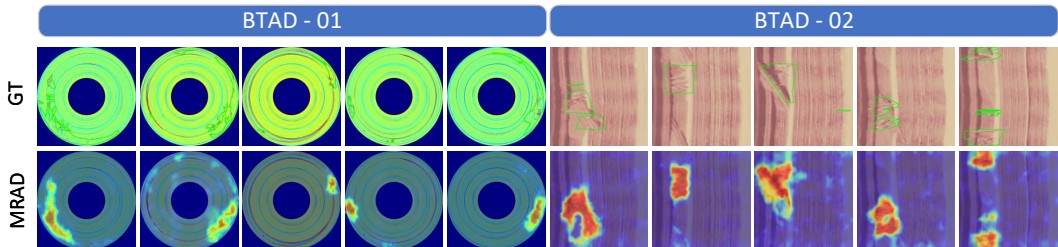

Figure 26: Anomaly maps of MRAD-CLIP across various categories. The first row shows the ground-truth annotations, and the second row shows the predicted anomaly maps.

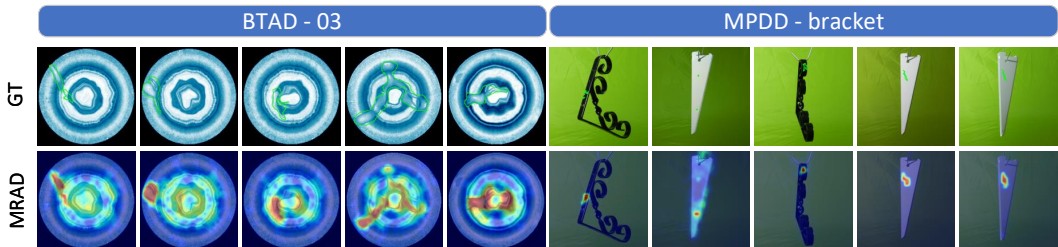

Figure 27: Anomaly maps of MRAD-CLIP across various categories. The first row shows the ground-truth annotations, and the second row shows the predicted anomaly maps.

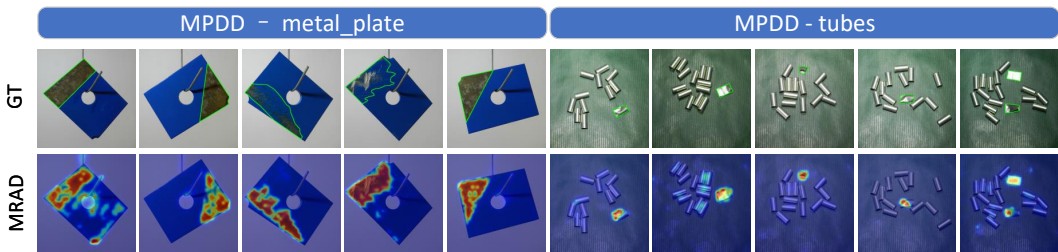

Figure 28: Anomaly maps of MRAD-CLIP across various categories. The first row shows the ground-truth annotations, and the second row shows the predicted anomaly maps.

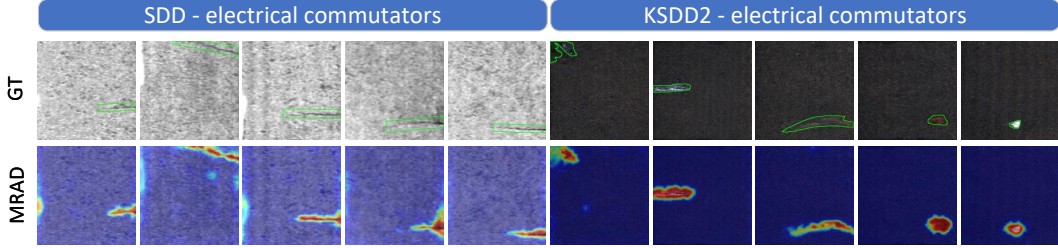

Figure 29: Anomaly maps of MRAD-CLIP across various categories. The first row shows the ground-truth annotations, and the second row shows the predicted anomaly maps.

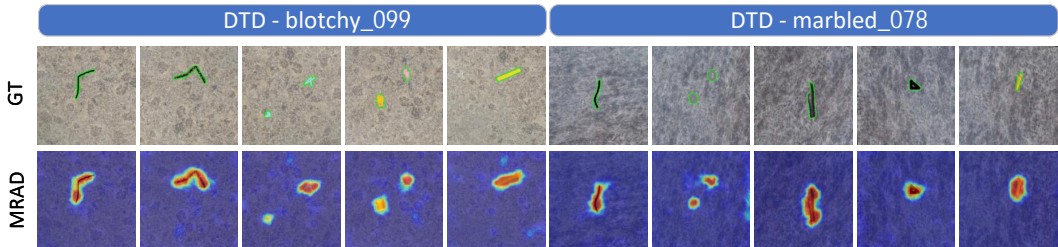

Figure 30: Anomaly maps of MRAD-CLIP across various categories. The first row shows the ground-truth annotations, and the second row shows the predicted anomaly maps.

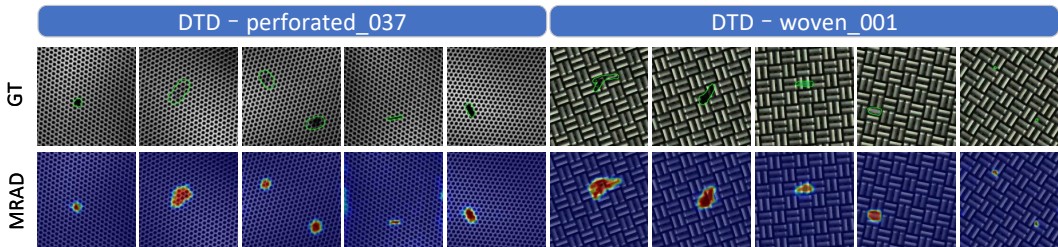

Figure 31: Anomaly maps of MRAD-CLIP across various categories. The first row shows the ground-truth annotations, and the second row shows the predicted anomaly maps.

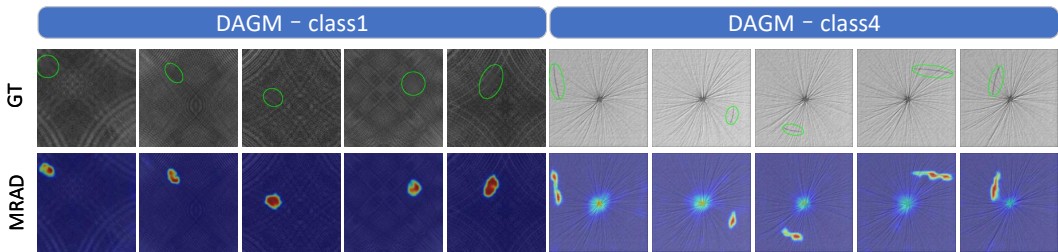

Figure 32: Anomaly maps of MRAD-CLIP across various categories. The first row shows the ground-truth annotations, and the second row shows the predicted anomaly maps.

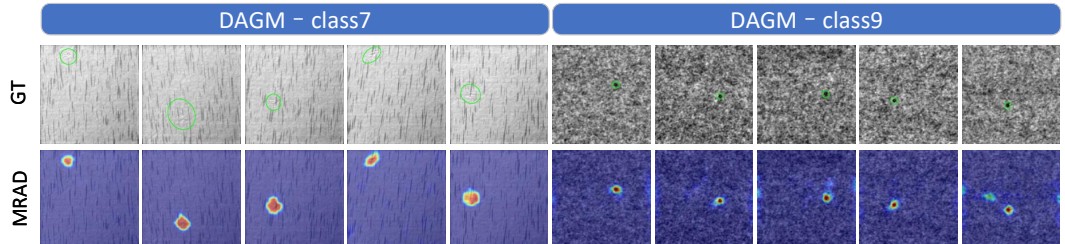

Figure 33: Anomaly maps of MRAD-CLIP across various categories. The first row shows the ground-truth annotations, and the second row shows the predicted anomaly maps.

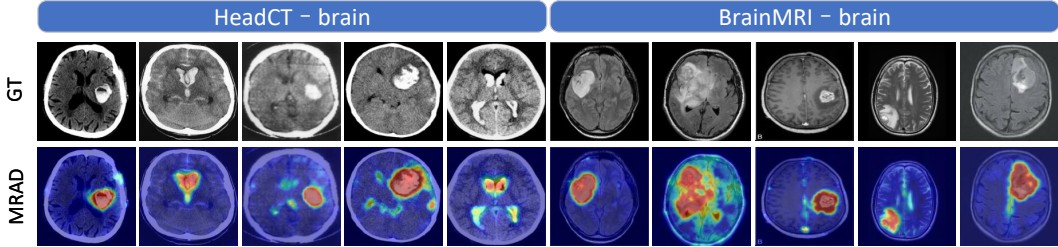

Figure 34: Anomaly maps of MRAD-CLIP across various categories. The first row shows the ground-truth annotations, and the second row shows the predicted anomaly maps.

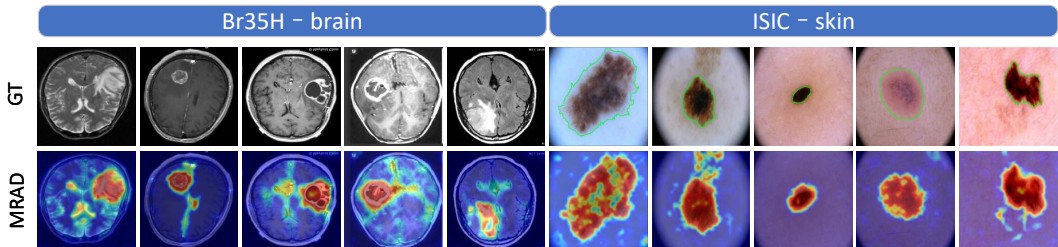

Figure 35: Anomaly maps of MRAD-CLIP across various categories. The first row shows the ground-truth annotations, and the second row shows the predicted anomaly maps.

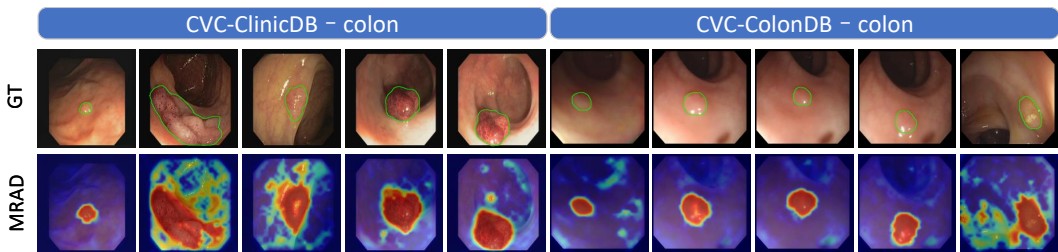

Figure 36: Anomaly maps of MRAD-CLIP across various categories. The first row shows the ground-truth annotations, and the second row shows the predicted anomaly maps.

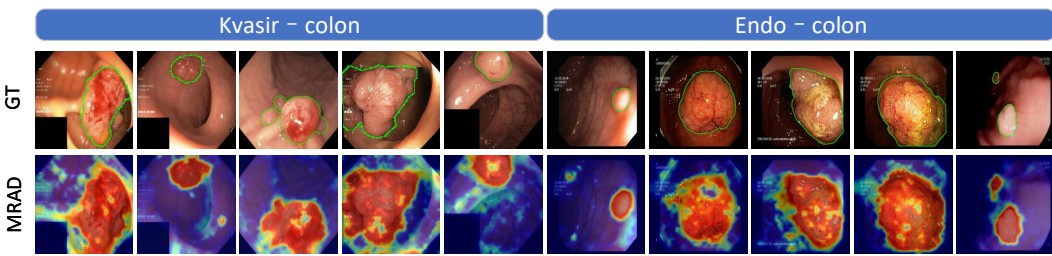

Figure 37: Anomaly maps of MRAD-CLIP across various categories. The first row shows the ground-truth annotations, and the second row shows the predicted anomaly maps.

