# OpenReview forum: "MRAD: Zero-Shot Anomaly Detection with Memory-Driven Retrieval"
_ICLR.cc/2026/Conference — ICLR 2026 Poster_

### Official Review · Reviewer_LY8o · 2025-10-31

**Soundness:** 3
**Presentation:** 3
**Contribution:** 3
**Rating:** 6
**Confidence:** 4

**Summary:**

This paper propose a two-level feature–label memory built from an auxiliary dataset using a frozen CLIP ViT-L/14. Inference is similarity retrieval from this memory (MRAD-TF). Here, MRAD-FT learns only two linear layers to calibrate the retrieval metric, while MRAD-CLIP injects region priors (normal/anomalous) from MRAD-FT as dynamic biases into learnable CLIP prompts to improve localization and cross-domain generalization. The approach is evaluated on 16 industrial/medical datasets and reports competitive train-free performance and new SOTA with the lightweight variants.

**Strengths:**

1.The paper rethinks zero-shot anomaly detection from a retrieval perspective, replacing complex prompt tuning or residual modeling with a memory-driven similarity framework.

2.The progression from MRAD-TF → MRAD-FT → MRAD-CLIP is logical and empirically validated. The fine-tuning stage (two linear layers + similarity dropout) improves separability while remaining lightweight, and the final CLIP-based variant integrates region-level priors as dynamic biases to guide attention and localization.

3.The authors evaluate across 16 datasets spanning industrial and medical domains, reporting both image- and pixel-level metrics. The consistent performance gains over prior CLIP-based ZSAD models (e.g., WinCLIP, AnomalyCLIP, FAPrompt) highlight strong generalization.

**Weaknesses:**

1.The approach depends on computing similarities against thousands of prototypes (≈3k here), yet the paper omits latency, memory, and scalability studies. As memory grows with more datasets or higher patch granularity, retrieval time could become a bottleneck.

2.Although the model is tested on diverse datasets, there is little reporting of variance (seed/template effects), robustness to distribution shifts (e.g., lighting, noise), or per-category breakdowns. The current results might reflect dataset bias or favorable template selection.

**Questions:**

1.What is the minimal supervision needed to construct the memory? Can MRAD-TF be instantiated with only normal data (no masks) + synthetic anomaly patches, and how would performance change? Any results with image-level labels only?

2.In Table 2 the class-token bias hurts performance.  Would multi-scale priors (coarse-to-fine) or attention-pooled region priors help?

---

> ### Author Response · Authors · 2025-11-19
> **Response to Reviewer LY8o (Part 1)**
>
> Thank you for your positive review and the insightful comments.
>
> **W1: The method depends on computing similarities against thousands of prototypes (about 3k in our setup), but the paper does not analyze latency, memory usage, or scalability. As the memory bank grows with more datasets or finer patch granularity, retrieval time could become a bottleneck.**
>
> A1: Thank you for this comment. Your concern is reasonable, since MRAD performs a $QK^\top$ similarity computation against a set of prototypes during retrieval, and in our current configuration the number of patch-level prototypes is around 3000. However, under the ViT-L/14-336 setting used in our experiments, this cost is practically manageable. The memory bank itself occupies only about 5 MB. For MRAD-FT, the peak training memory is around 7.8 GB and the trainable module is about 10 MB, and the model converges in 1 epoch on a single RTX 3090 with an epoch time of about 334 seconds. Both memory usage and training time are not higher than those of CLIP prompt-learning baselines such as AnomalyCLIP, FAPrompt, and AdaCLIP, whose additional parameters range from about 22 MB to 41 MB, with peak memory around 7.9–12 GB and per-epoch training times around 341–753 seconds. On the inference side, we also profiled MRAD-FT in detail: the average total inference time per image is about 198 ms, where image-level and pixel-level retrieval together account for about 110 ms and the rest comes from backbone forward pass and post-processing. We consider this overhead acceptable for our setting. A complete comparison of parameter scales and training resources has been added in **Appendix C.3**.
>
> Regarding scalability when the data volume increases, this is indeed an important issue. For this reason, we analyzed memory size in both the main paper (Sec. 4.3, *Memory size ablation*) and additional experiments. Simple random subsampling of the memory bank leads to a slight performance drop. In contrast, when we apply clustering-based compression (for example, clustering the prototypes into 100 clusters and using cluster centers as new prototypes), the number of prototypes can be reduced from about 3000 to 100, and MRAD-FT/MRAD-CLIP on MVTec still achieve P-AUROC / I-AUROC of about 92.2 / 92.0 and 92.8 / 93.8, which is very close to the full-memory setting. This suggests that when the auxiliary data grow larger or a finer patch granularity is needed, we can control the number of prototypes at the scale of a few hundred through clustering-based compression or similar techniques and thus keep both similarity computation time and memory usage within a reasonable range while essentially preserving performance.

---

> ### Author Response · Authors · 2025-11-19
> **Response to Reviewer LY8o (Part 2)**
>
> **W2: Although the model is tested on diverse datasets, there is little reporting of variance (seed/template effects), robustness to distribution shifts (e.g., lighting, noise), or per-category breakdowns. The current results might reflect dataset bias or favorable template selection.**
>
> A2: Thank you for this suggestion. We agree that examining the robustness and stability of the method is very important.
>
> First, regarding randomness, we performed multiple runs with different seeds for MRAD-FT and MRAD-CLIP on MVTec. The results are summarized below:
>
> | Method     | P-AUROC (mean) | I-AUROC (mean) | Max deviation |
> |-----------|-----------------|----------------|---------------|
> | MRAD-FT   | 92.2            | 91.9           | ≤ 0.5         |
> | MRAD-CLIP | 92.7            | 93.6           | ≤ 0.4         |
>
> The overall fluctuations are within ±0.5%, and the gap between these averages and the best results reported in the paper is only around 0.3 percentage points. In addition, we did not cherry-pick seeds to report better numbers, and the seed settings are kept consistent with baselines such as AnomalyCLIP and FAPrompt.
>
> Second, for robustness under distribution shifts, we applied mild test-time perturbations to the MVTec test images, including random brightness changes of up to ±15% and additive Gaussian noise with standard deviation 5, and then re-evaluated the models on the perturbed set. The results are:
>
> | Method     | P-AUROC (orig.) | I-AUROC (orig.) | P-AUROC (noisy+bright.) | I-AUROC (noisy+bright.) |
> |-----------|------------------|------------------|--------------------------|--------------------------|
> | MRAD-FT   | 92.2             | 92.3             | 91.6                     | 91.1                     |
> | MRAD-CLIP | 93.0             | 94.0             | 92.6                     | 92.8                     |
>
> Both MRAD-FT and MRAD-CLIP show only slight performance drops, with decreases in P-AUROC and I-AUROC below 1 percentage point. MRAD-CLIP is slightly more robust overall and still outperforms AnomalyCLIP and FAPrompt on the original MVTec setting. These results indicate that the model maintains reasonable robustness to typical illumination and noise shifts and does not collapse under moderate degradation of image quality.
>
> Finally, regarding per-category breakdowns, the updated **Appendix C.9** now reports per-category results for MVTec and VisA as two representative datasets. We can provide per-category results for additional datasets if needed.

---

> ### Author Response · Authors · 2025-11-19
> **Response to Reviewer LY8o (Part 3)**
>
> **Q1.1: What is the minimal supervision needed to construct the memory? Are there any results with image-level labels only?**
>
> A1.1: In terms of supervision, if we aim to perform both image-level detection and pixel-level segmentation, MRAD requires at least pixel-level masks to construct patch-level memory prototypes. The motivation for our two-level memory design follows the division of roles in ViT, where the class token focuses on global semantic information and patch tokens encode local structure. Accordingly, we store class tokens in an image-level memory primarily for image-level retrieval and classification, while patch tokens are stored in a pixel-level memory for segmentation and localization. The segmentation output can then be aggregated via Top-K pooling to provide a complementary image-level anomaly score, whereas a pure classification branch cannot provide any guidance back to the pixel level.
>
> If only image-level labels are available and pixel-level memory cannot be constructed, we can still retain the image-level memory and obtain classification scores through prototype retrieval at the image level. However, in this case there is no segmentation branch or Top-K aggregation path, and the model cannot produce any pixel-level localization. Without this complementary branch, image-level performance degrades. For example, the I-AUROC of MRAD-FT on MVTec drops from 92.3 to around 90.0, and on VisA from 85.5 to about 83.7, which is clearly weaker than the full two-level memory configuration. In addition, we also report in the appendix the effect of removing the image-level memory on classification performance, where we observe varying degrees of degradation across different datasets. Detailed results are provided in **Appendix C.4** (“Ablation on two-level memory bank”).
>
> **Q1.2: Whether MRAD can be instantiated with only normal data (no masks) plus synthetic anomaly patches, and how the performance would change.**
>
> A1.2: Thank you for this very inspiring suggestion. We agree that this is a meaningful direction and therefore conducted dedicated experiments, with detailed results reported in **Appendix C.8**.
>
> In this experiment, we use VisA as the auxiliary dataset to construct normal prototypes and apply DRAEM-style local perturbations (Cut&Paste) on normal images to synthesize anomalous regions. These synthetic anomalies are then used in place of real anomalies to build anomaly prototypes, while keeping all other settings unchanged, and we evaluate ZSAD on MVTec. The main results are:
>
> | Method    | Anomaly source     | P-AUROC | I-AUROC |
> |-----------|--------------------|---------|---------|
> | MRAD-FT   | Real anomalies     | 92.2    | 92.3    |
> |           | Synthetic patches  | 89.2    | 79.7    |
> | MRAD-CLIP | Real anomalies     | 93.0    | 94.0    |
> |           | Synthetic patches  | 83.1    | 76.1    |
>
> These results show that under this setting the MRAD framework still functions, but overall performance clearly degrades compared with using real anomalies. For MRAD-FT, P-AUROC / I-AUROC are about 89.2 / 79.7, corresponding to drops of around 3% and 10%, although the segmentation performance remains reasonably good. For MRAD-CLIP, the performance is about 83.1 / 76.1, with a larger drop exceeding 10%. By examining the loss curves, we observe that CLIP-based prompt-learning is prone to overfitting to this DRAEM-style synthetic anomaly pattern, which is a non-natural form of anomaly, and this tendency aligns with limitations observed in other CLIP-based ZSAD methods.
>
> Overall, these findings are consistent with our expectations. The retrieval mechanism of MRAD relies on the anomaly prototypes in the memory bank covering real anomaly patterns, whereas synthetic anomalies mainly simulate local texture perturbations and lack the semantic diversity of real industrial defects. As a result, their coverage is insufficient for robust cross-class retrieval. At the same time, this “normal data + synthetic anomalies” regime is still informative: for example, under such weak supervision, MRAD-FT maintains reasonably good segmentation performance, which suggests that a memory-based retrieval framework retains potential extensibility in settings without real anomalies and further highlights the advantages of retrieval-based modeling.
>
> **Reference**
>
> [1] Vitjan Zavrtanik, Matej Kristan, and Danijel Skočaj. *DRAEM: A discriminatively trained reconstruction embedding for surface anomaly detection*. In **Proceedings of the IEEE/CVF International Conference on Computer Vision (ICCV)**, pp. 8330–8339, 2021.

---

> ### Author Response · Authors · 2025-11-19
> **Response to Reviewer LY8o (Part 4)**
>
> **Q2: In Table 2, the class-token bias hurts performance. Would multi-scale priors (coarse-to-fine) or attention-pooled region priors help?**
>
> A2: Thank you for this insightful question. We agree with the observation that Table 2 clearly shows class-token bias significantly hurts performance. The main reason is that the class token encodes only global semantics and lacks spatial resolution, so it cannot reliably indicate where anomalies are located.
>
> Regarding multi-scale priors, in early versions of our implementation we did try to extract coarse-to-fine patch tokens from different ViT layers (for example, layers 6, 12, 18, and 24) and use them for alignment or retrieval. We then stacked or concatenated these intermediate features and injected them as biases. Under the same training and evaluation settings, however, this multi-layer combination did not bring consistent improvements and in some configurations even led to slight degradation. For this reason, we ultimately adopted a simpler and more stable single-scale design.
>
> For attention-pooled region priors, our experimental findings are consistent with your intuition. Table 2 in Section 4.3 (“Ablation on dynamic prompt biases”) shows that the cross-patch bias, where two learnable queries attend to patch features, clearly outperforms the class-token bias on all datasets. This indicates that focusing the bias on local patch regions rather than on a global class token is beneficial. Building on this idea, in the final version we use region priors provided by MRAD-FT. Concretely, MRAD-FT estimates an anomaly confidence for each patch, and we perform weighted average pooling over high-confidence regions to obtain an “anomalous region embedding”, which is then injected into CLIP prompts as a dynamic bias. As shown in Table 2, both the anomaly-only prior and the dual prior that combines normal and anomalous regions achieve near-optimal performance at both the pixel and image levels. Intuitively, this region prior serves as a spatial alignment signal for the “normal” and “anomalous” prompt embeddings, enhancing cosine similarity in the corresponding areas and enabling more accurate matching of anomalous regions.
>
> In summary, our experiments indicate that simply stacking multi-layer intermediate features does not reliably improve performance. In contrast, focusing attention on high-confidence anomalous patches and pooling them into an explicit prior embedding to drive dynamic prompts is currently the most effective and stable bias design within our framework.

---

### Official Review · Reviewer_toN8 · 2025-11-01

**Soundness:** 3
**Presentation:** 3
**Contribution:** 3
**Rating:** 6
**Confidence:** 4

**Summary:**

This paper presents a novel method for ZSAD called Memory-Retrieval Anomaly Detection (MRAD). It replaces parametric fitting with with direct memory retrieval, facilitating anomaly detection without extensive training. The framework constructs a two-level memory bank using feature-label pairs, allowing efficient similarity retrieval and demonstrating superior performance. The work highlights the potential of leveraging the empirical data distribution for effective anomaly detection, offering a fresh perspective on ZSAD.

**Strengths:**

This paper proposes a novel approach that replaces parametric fitting with a direct memory retrieval to ZSAD, offering a fresh perspective on anomaly detection. It demonstrates soundness in both theoretical grounding and empirical validation. Also, the paper gives clear definitions and explanations of methodologies, making it accessible to readers.

**Weaknesses:**

Major:
1. All experiments use VisA or MVTec-AD as the auxiliary dataset. Could other datasets be used as the auxiliary dataset?
2. MRAD-CLIP injects region priors as additive biases into CLIP’s learnable prompts. It remains unclear whether this choice of design is optimal or merely sufficient.
3. MRAD-FT adds 2.76M parameters, but the fine-tuning efficiency remains under-explored. This may result in the inadequately quantified “lightweight” claim.

Minor:
1. Memory bank size scales with the auxiliary training data, which could be optimized further.
2. The number of the medical datasets used for image-level ZSAD is relatively small.

**Questions:**

1. Is there any related work about similarity retrieval in the field of ZSAD before?
2. Are the quality and diversity of the feature-label pairs stored in the memory bank needed to be controlled or ensured?
3. What are the known limitations of MRAD, particularly in scenarios with highly imbalanced datasets or extreme domain shifts?

---

> ### Author Response · Authors · 2025-11-19
> **Response to Reviewer toN8 (Part 1)**
>
> Thank you very much for your comments.
>
> **W1: All experiments use VisA or MVTec-AD as the auxiliary dataset. Could other datasets be used as the auxiliary dataset?**
>
> A1: Thank you for this comment. In most existing ZSAD works, the generalization ability is usually evaluated by default with MVTec-AD or VisA as the training or auxiliary dataset, and there is very little systematic analysis of what happens when the auxiliary source is changed. This makes your point particularly important: if experiments are only conducted on these two standard datasets, it is hard to tell whether a method implicitly depends on a specific data source.
>
> With this in mind, we conducted a further validation for MRAD. In addition to using VisA and MVTec-AD as auxiliary datasets, we also selected several other industrial and medical datasets and used each of them as the only auxiliary source, that is, we constructed the memory and trained MRAD solely on that dataset and then evaluated zero-shot performance on MVTec-AD in a unified protocol. **The complete results, analysis, and dataset statistics are reported in Appendix C.5.** The main trends are summarized below (here P and I denote pixel-level and image-level AUROC, respectively):
>
> | Domain     | Aux. dataset  | Classes | Normal | Anomaly | MRAD-FT (P / I) | MRAD-CLIP (P / I) |
> |-----------|---------------|---------|--------|---------|-----------------|-------------------|
> | Industrial| VisA          | 12      | 962    | 1200    | 92.2 / 92.3     | 93.0 / 94.0       |
> |           | BTAD          | 3       | 451    | 290     | 92.0 / 92.5     | 91.0 / 92.2       |
> |           | MPDD          | 6       | 176    | 282     | 91.1 / 87.3     | 91.2 / 89.2       |
> |           | DTD-Synthetic | 12      | 357    | 947     | 91.3 / 88.7     | 91.5 / 89.8       |
> |           | SDD           | 1       | 181    | 74      | 88.1 / 82.5     | 88.9 / 87.8       |
> |           | KSDD2         | 1       | 894    | 110     | 90.2 / 86.3     | 92.1 / 92.8       |
> | Medical   | CVC-ClinicDB  | 1       | 0      | 612     | 87.5 / 75.0     | 88.1 / 79.7       |
> |           | Kvasir        | 1       | 0      | 1000    | 86.0 / 70.6     | 84.6 / 73.1       |
>
> These experiments show that MRAD is not tied to using VisA or MVTec specifically, but to whether the auxiliary dataset provides enough diversity for learning a cross-class anomaly pattern that supports retrieval. Since each query retrieves similarities against prototypes from multiple classes, the auxiliary dataset must supply sufficiently varied normal and anomalous patterns for a class-agnostic anomaly representation to form.
>
> Concretely, when the auxiliary dataset comes from the industrial domain and has a reasonable number of classes or diverse defect shapes (such as VisA, BTAD, MPDD, and DTD-Synthetic), the prototype distribution in feature space covers variations across multiple classes. During training, the model repeatedly “uses anomalies from one class to retrieve against prototypes of other classes”, which encourages it to form a transferable cross-class anomaly concept. Under these auxiliary datasets, MRAD shows relatively stable cross-dataset performance, and its pixel-level metrics are generally stronger than baselines such as AnomalyCLIP. The variation on image-level metrics is mainly due to the limited semantic expressiveness of the class token on some auxiliary datasets. For example, DTD has multiple nominal “classes”, but in practice they share very similar textures and backgrounds, which leads to limited semantic coverage in the image-level memory.
>
> In contrast, when the auxiliary dataset is essentially single-class, such as SDD or KSDD2, the model has almost no opportunity to learn true cross-class retrieval during training. The anomaly prototypes then only capture a specific defect morphology, and it is difficult to form a transferable anomaly pattern. As a result, the cross-domain performance of MRAD-FT drops more noticeably. In the more extreme case where the auxiliary dataset is both single-class and strongly out-of-domain, such as medical datasets Kvasir and ClinicDB, the prototype distribution differs markedly from industrial images. There is no meaningful cross-class contrast and no anomaly patterns aligned with the target domain, which makes cross-domain retrieval particularly difficult and leads to the most severe degradation.
>
> Overall, these results indicate that MRAD does not rely on any particular dataset as the auxiliary source. As long as the auxiliary dataset lies in a related domain and provides sufficient category and appearance diversity, the model can learn a transferable cross-class anomaly retrieval ability during training and maintain relatively stable performance in cross-dataset evaluation. Significant degradation mainly appears when the auxiliary dataset has very limited semantic coverage or a large domain shift relative to the target, but even in these extreme cases the model does not completely fail.

---

> ### Author Response · Authors · 2025-11-19
> **Response to Reviewer toN8 (Part 2)**
>
> **W2: Whether injecting region priors as additive biases into CLIP’s learnable prompts is an optimal design choice or merely a sufficient one.**
>
> A2: Thank you for this question. In our current framework, using region priors as dynamic additive biases is a simple and effective choice. Conceptually, this follows the general consensus in recent prompt-learning literature that dynamic prompts tend to generalize better than static ones. Based on this, we conducted a set of ablations on the source of the bias, as reported in Section 4.3 (“Ablation on dynamic prompt biases”) and Table 2. These ablations cover several commonly used alternatives in existing ZSAD works, including static prompts without any bias (equivalent to AnomalyCLIP), using only the class token as a global bias, and using cross-attention–derived regional context as the bias. When integrating MRAD-FT, we further explored two variants that inject only the anomaly prior or inject both the normal and anomaly priors.
>
> The empirical results indicate that pure global biases such as class tokens weaken segmentation performance and cross-domain robustness, suggesting that global features alone are insufficient. Introducing cross-patch regional biases improves both image-level and pixel-level metrics. Building on this, injecting region priors obtained from MRAD-FT, either anomaly-only or normal-plus-anomaly, consistently yields the best or most stable performance across almost all datasets. Intuitively, these region priors act as a spatial alignment signal for the “normal” and “anomalous” prompt embeddings, enhancing local cosine similarity in the corresponding areas and enabling more accurate matching of anomalous regions.
>
> Based on both empirical evidence and this intuition, we believe that injecting region priors as dynamic biases is an effective and well-justified design choice within the current framework.
>
> **W3: MRAD-FT adds 2.76M parameters, but the fine-tuning efficiency is under-explored, which may make the “lightweight” claim insufficiently quantified.**
>
> A3: Thank you for this suggestion. Our notion of “lightweight fine-tuning” refers not only to the small number of additional parameters, but also to the actual training cost. Under the same ViT-L/14 backbone and MVTec-AD setting, MRAD-FT introduces only about 2.76M extra trainable parameters, corresponding to a model size of roughly 10 MB, and the memory bank itself is only about 2–5 MB. The peak training memory usage is around 7.8 GB, and the model converges in just one epoch on a single RTX 3090 GPU, with each epoch taking about 334 seconds.
>
> In comparison, the trainable modules of AnomalyCLIP, FAPrompt, and AdaCLIP are larger, with model sizes of about 22 MB, 39 MB, and 41 MB respectively, peak training memory of roughly 7.9, 11, and 12 GB, and they typically require 10–15 epochs to converge. Their per-epoch training times are also longer (about 341 s, 449 s, and 753 s). Despite the significantly lower parameter count and training cost, MRAD-FT still achieves the best or near-best performance among these methods at both the image and pixel levels. We have added a complete comparison of parameter scales and training resources in **Appendix C.3** to quantitatively support our claim of “lightweight” fine-tuning.

---

> ### Author Response · Authors · 2025-11-19
> **Response to Reviewer toN8 (Part 3)**
>
> **W4: The size of the memory bank scales with the amount of auxiliary training data, and could potentially be further optimized.**
>
> A4: Thank you for this comment. As you noted, since retrieval involves a $QK^\top$ operation, the size of $K$ grows linearly with the number of patch-level prototypes, and thus the memory bank size scales linearly with the volume of auxiliary data. This is indeed an important aspect to consider. In the initial version of the paper, we already reported a “memory size ablation” in the main text (Sec. 4.3, *Memory size ablation*), where we randomly subsampled the memory bank. As the memory size is reduced, performance drops slightly. The underlying reason is that random subsampling cannot guarantee sufficient coverage and stability of the prototype distribution, so the resulting memory is less uniform and less representative than the full memory.
>
> Building on this, we conducted a more targeted experiment. We first cluster the features in the memory bank, for example into 100 clusters, and then use the cluster centers as new memory prototypes, fixing the total number of entries to 100. On MVTec, we observe that both MRAD-FT and MRAD-CLIP achieve performance that is very close to and stable compared with the full-memory setting, with P-AUROC / I-AUROC of about 92.2 / 92.0 and 92.8 / 93.8 respectively, which corresponds to only around a 0.2 percentage point drop. This indicates that clustering-based compression can substantially reduce the memory size while essentially preserving performance.
>
> At the same time, we would like to emphasize that under the experimental scale considered in the paper, keeping all training samples as memory is feasible and consistent with our design principle of preserving as much of the empirical distribution as possible. Under this setting, GPU memory usage and inference latency remain within a reasonable range. However, when the auxiliary training data grows to tens of thousands of samples, we recommend using clustering-based compression or similar techniques to further optimize memory-bank size and retrieval efficiency while maintaining accuracy.
>
> **W5: The number of medical datasets used for image-level ZSAD is relatively small.**
>
> A5: Thank you for this comment. At present, publicly available medical datasets that both provide reliable image-level labels and are suitable for the zero-shot anomaly detection (ZSAD) setting are indeed very limited. As a result, most existing ZSAD works that consider image-level evaluation in the medical domain also focus on only a small number of benchmark datasets.
>
> To further assess the applicability of our method in medical scenarios, we additionally organized a Liver CT dataset containing 1,493 CT slices with image-level labels from the BMAD benchmark. The experimental setup and results on this dataset have been added to **Appendix C.6.** The results show that on this new medical ZSAD benchmark, our method still achieves relatively strong performance compared with other representative approaches, which provides additional evidence for the generalization ability of MRAD in medical imaging settings.
>
> **Reference**
>
> [1] Bao, J., Sun, H., Deng, H., He, Y., Zhang, Z., & Li, X. BMAD: Benchmarks for Medical Anomaly Detection. arXiv:2306.11876, 2024.

---

> ### Author Response · Authors · 2025-11-19
> **Response to Reviewer toN8 (Part 4)**
>
> **Q1: Whether there is any prior related work on similarity-based retrieval in the ZSAD setting.**
>
> A1: Regarding similarity-based retrieval, most existing zero-shot anomaly detection methods are still primarily CLIP-based prompt-learning approaches. They typically optimize “normal/abnormal” text prompts to align with anomalous regions in images, as in AnomalyCLIP, AdaCLIP, and related works. To the best of our knowledge, within the ZSAD setting there is no prior work that explicitly constructs a feature–label memory and uses cross-image, cross-class similarity retrieval as the main decision mechanism in the way MRAD does.
>
> **Q2: Whether the quality and diversity of the feature–label pairs stored in the memory bank need to be controlled or ensured.**
>
> A2: Thank you for this question. The quality and diversity of the feature–label pairs in the memory bank do indeed affect the performance of MRAD, and therefore they need to be controlled and ensured. We observe this from two sets of experiments.
>
> First, in the “changing auxiliary dataset” experiments, when the auxiliary dataset has rich categories and diverse appearances, MRAD-FT and MRAD-CLIP show only minor drops on MVTec compared with using VisA as the auxiliary source. In contrast, when the auxiliary dataset is essentially single-class, consists mainly of texture defects, or comes from a domain that is very different from industrial imagery such as medical datasets, the performance degrades more noticeably. This indicates that the memory bank must contain sufficiently diverse semantic features and visual patterns in order to support cross-class anomaly retrieval.
>
> Second, in the “memory size and compression” experiments, we find that naive random subsampling of the memory causes a slight performance drop. However, when we first cluster the memory features and then use cluster centers as new prototypes, even after compressing the number of prototypes down to 100 entries, the performance on MVTec remains very close to that of the full-memory setting. This suggests that as long as distributional coverage is preserved, MRAD is not overly sensitive to the exact memory size.
>
> Taken together, these findings suggest that it is important to prioritize the diversity and label quality of the auxiliary dataset and to maintain a representative memory bank. Under these conditions, the model can better acquire the ability to perform cross-class retrieval of a shared anomaly pattern.
>
> **Q3: The known limitations of MRAD, especially under highly imbalanced datasets or extreme domain shifts.**
>
> A3: Thank you for raising this important point. As discussed in the main paper, the key limitation of MRAD comes from its dependency on the quality and diversity of the feature–label pairs stored in the memory bank. Since MRAD performs zero-shot anomaly detection through cross-class retrieval, the auxiliary dataset must provide sufficiently diverse normal and abnormal prototypes so that the model can learn a shared, class-agnostic anomaly pattern. When this condition is not met, performance degradation is expected.
>
> The limitation becomes clear in scenarios with severe class imbalance. For example, in single-class industrial datasets such as KSDD2 or SDD—where KSDD2 further has a strong imbalance between normal and abnormal samples—the anomaly prototypes are sparse and statistically unstable. When these datasets are used as the auxiliary source, the memory bank lacks adequate diversity, leading to noticeable drops in cross-domain performance for MRAD-FT. MRAD-CLIP is slightly more robust in such cases, but the overall trend confirms that insufficient anomaly coverage constrains the effectiveness of retrieval-based ZSAD.
>
> A similar limitation appears under extreme domain shifts. When the auxiliary data come from a domain that is very different from industrial imagery—such as the medical datasets Kvasir or ClinicDB—the semantic and texture statistics of normal and abnormal regions deviate significantly from those in the target domain. Under such strong domain mismatch, both MRAD-FT and MRAD-CLIP exhibit clear performance degradation. This aligns with our findings in the auxiliary-dataset experiments: when the auxiliary set does not provide semantically compatible or visually aligned prototypes, the learned anomaly pattern does not transfer, and retrieval becomes unreliable.
>
> Overall, MRAD is most effective when the auxiliary dataset offers sufficient category or appearance diversity within a broadly aligned domain. Its main limitations emerge when the memory bank is built from datasets that are extremely imbalanced or severely out-of-domain, where it becomes difficult for the model to acquire a transferable, class-agnostic anomaly pattern.

---

### Official Review · Reviewer_iQze · 2025-11-01

**Soundness:** 3
**Presentation:** 3
**Contribution:** 2
**Rating:** 6
**Confidence:** 3

**Summary:**

This paper proposes MRAD, a unified framework for zero-shot anomaly detection (ZSAD) that replaces parametric modeling with direct memory-based retrieval. The method freezes the CLIP image encoder and builds a two-level memory bank (image-level and pixel-level) from auxiliary data. The MRAD-TF variant operates in a fully training-free manner and already achieves competitive results through similarity-based retrieval. The MRAD-CLIP variant further injects normal and anomalous region priors (derived from MRAD-FT) into learnable text prompts. Experiments on 16 industrial and medical datasets demonstrate the effectiveness of the approach.

**Strengths:**

1. The framework is simple and effective.

2. The paper is clearly written and easy to follow.

3. Extensive experiments on both industrial and medical benchmarks support the claims.

**Weaknesses:**

1. The fine-tuning stage adopts two linear projection layers, but no ablation compares against shallower (e.g., 1-layer) or deeper variants, making it unclear whether the chosen depth is optimal or arbitrary.

2. The method emphasizes the benefit of two-level memory (image + pixel), but there is no ablation where one level is removed to show whether both levels are truly necessary.

3. No sensitivity analysis is provided for key hyperparameters (e.g., similarity mask ratio ρ, top-k selection, thresholding strategy).

**Questions:**

1. The fine-tuning stage adopts two linear projection layers, but no ablation compares against shallower (e.g., 1-layer) or deeper variants, making it unclear whether the chosen depth is optimal or arbitrary.

2. The method emphasizes the benefit of two-level memory (image + pixel), but there is no ablation where one level is removed to show whether both levels are truly necessary.

3. No sensitivity analysis is provided for key hyperparameters (e.g., similarity mask ratio ρ, top-k selection, thresholding strategy).

4. The approach relies on a specific auxiliary dataset, yet there is no experiment showing whether the performance is stable when using different auxiliary datasets.

**Details Of Ethics Concerns:**

No ethics concerns.

---

> ### Author Response · Authors · 2025-11-19
> **Response to Reviewer iQze (Part 1)**
>
> Thank you very much for your comments.
>
> **Q1: Whether the choice of using two linear projection layers in the fine-tuning stage is optimal, given that no ablation compares against shallower (e.g., 1-layer) or deeper variants.**
>
> A1: Thank you for this comment. Our use of “two linear projections” does not refer to a two-layer deep network, but rather to applying one linear mapping to Q and one to K respectively, following the standard design of independent $W_Q$ and $W_K$ in attention mechanisms.
>
> To address the concern about projection depth, we conducted an additional controlled experiment on MVTec: keeping all other settings identical, we varied the projection depth from 1 to 4 layers. The resulting P-AUROC scores are summarized below:
>
> | Projection depth  | 1    | 2    | 3    | 4    |
> |---------------------------|------|------|------|------|
> | P-AUROC (%)               | 92.2 | 90.8 | 88.5 | 61.3 |
>
> These results show that a single-layer linear mapping performs best, while deeper variants lead to clear degradation due to overfitting. This empirically supports our current design choice.
>
> **Q2: Whether both levels of the proposed two-level memory (image-level and pixel-level) are truly necessary, given that no ablation removes one level.**
>
> A2: Thank you for this suggestion. Our design of the two-level memory follows the division of roles in ViT: the class token primarily encodes global semantic information, while patch tokens focus on local structure and spatial details. Accordingly, MRAD stores the image-level class token in an image-level memory for classification, and stores patch tokens in a pixel-level memory for segmentation and localization.
>
> When the pixel-level memory is removed, patch-level retrieval becomes impossible, and the model loses its localization capability entirely, leading to no meaningful pixel-level segmentation. On the image-level side, the absence of the segmentation branch and its Top-K aggregation also causes performance degradation. For example, MRAD-FT’s I-AUROC drops from 92.3 to around 90.0 on MVTec, and from 85.5 to about 83.7 on VisA—both noticeably worse than the full two-level configuration.
>
> Conversely, when only the pixel-level memory is kept and the image-level anomaly score is only obtained via Top-K aggregation of pixel responses, localization remains intact but image-level classification deteriorates. Across most datasets, I-AUROC drops to varying degrees, such as from 92.3 to 91.7 on MVTec and from 85.5 to 79.4 on VisA.
>
>
> In summary, pixel-level memory is essential for anomaly localization, while robust image-level detection relies on the complementary contributions of both image- and pixel-level memories. For the overall framework, the two levels are jointly necessary. **More complete results are provided in Appendix C.4 (“Ablation on two-level memory bank”).**
>
> **Q3: Lack of sensitivity analysis for key hyperparameters such as the similarity mask ratio ρ, the top-K selection, and the thresholding strategy.**
>
> A3: Thank you for pointing this out. The sensitivity analysis for the similarity-mask ratio ρ has already been included in Appendix C.4 (“Ablation on mask threshold”), where detailed results are provided. Overall, when the masking strategy is used, the performance differences across different ρ values are small, indicating that the method is not highly sensitive to this parameter.
>
> In addition, we have added an ablation study on the Top-K selection, where K denotes the percentage of pixels retained relative to the total number of pixels in the image. On MVTec, we get the results:
>
> | K (ratio of pixels) | 0.0001 | 0.001 | 0.01 | 0.1  |
> |---------------------|--------|-------|------|------|
> | I-AUROC (%)         | 93.6  | 93.9 | 94.0| 92.0|
>
> Performance decreases at both extremely small and excessively large K values, while our chosen value (0.01) lies in a relatively stable region, consistent with our empirical expectation of typical small-defect sizes.
>
> For the threshold used to extract the anomaly prior, we adopt a default value of 0.5 and conducted multiple comparisons around this setting. The results show that P-AUROC varies only slightly within the mid-range near 0.5 (roughly 92.9–93.2), suggesting that the overall performance is not sensitive to this threshold. This further demonstrates the stability and robustness of our framework with respect to hyperparameter choices.

---

> ### Author Response · Authors · 2025-11-19
> **Response to Reviewer iQze (Part 2)**
>
> **Q4: Whether the approach relies on a specific auxiliary dataset, and whether performance remains stable when different auxiliary datasets are used.**
>
> A4: Thank you for raising this point. It is worth noting that most existing ZSAD works typically default to using MVTec-AD or VisA as the training or auxiliary dataset, and very few studies systematically examine how changing the auxiliary source affects model performance. As a result, an important question has remained insufficiently explored in prior work: whether a method implicitly depends on a particular dataset. We fully agree with this concern and therefore conducted a dedicated set of experiments: for multiple candidate datasets, we used each one as the *only* auxiliary source (constructing the memory and training solely on that dataset), and then evaluated zero-shot detection on MVTec-AD. **A complete comparison table, analysis, and dataset statistics are provided in Appendix C.5.** The main trends are summarized below (here P and I denote pixel-level and image-level AUROC, respectively):
>
> | Domain     | Aux. dataset  | Classes | Normal | Anomaly | MRAD-FT (P / I) | MRAD-CLIP (P / I) |
> |-----------|---------------|---------|--------|---------|-----------------|-------------------|
> | Industrial| VisA          | 12      | 962    | 1200    | 92.2 / 92.3     | 93.0 / 94.0       |
> |           | BTAD          | 3       | 451    | 290     | 92.0 / 92.5     | 91.0 / 92.2       |
> |           | MPDD          | 6       | 176    | 282     | 91.1 / 87.3     | 91.2 / 89.2       |
> |           | DTD-Synthetic | 12      | 357    | 947     | 91.3 / 88.7     | 91.5 / 89.8       |
> |           | SDD           | 1       | 181    | 74      | 88.1 / 82.5     | 88.9 / 87.8       |
> |           | KSDD2         | 1       | 894    | 110     | 90.2 / 86.3     | 92.1 / 92.8       |
> | Medical   | CVC-ClinicDB  | 1       | 0      | 612     | 87.5 / 75.0     | 88.1 / 79.7       |
> |           | Kvasir        | 1       | 0      | 1000    | 86.0 / 70.6     | 84.6 / 73.1       |
>
> Whether MRAD can maintain performance across datasets fundamentally depends on whether the auxiliary source enables the model to learn a *class-agnostic anomaly pattern* that supports cross-class retrieval. Since MRAD requires each query to retrieve across prototypes from multiple classes during training, the auxiliary dataset must contain sufficiently diverse normal and anomalous patterns for such a class-independent anomaly pattern to form in the source domain.
>
> In the industrial domain, as long as the auxiliary dataset provides adequate category or defect-shape diversity (e.g., VisA, BTAD, MPDD, DTD-Synthetic), the prototype distribution covers variations across multiple classes. This allows the model to repeatedly “use anomalies from class A to retrieve against classes B and C” during training, thereby learning a consistent cross-class anomaly concept. As shown in the table, these datasets yield relatively stable zero-shot performance, and the pixel-level results consistently outperform baselines such as AnomalyCLIP (91.1). Image-level performance shows larger fluctuations primarily because the class token is more affected by global image quality, and its semantic expressiveness is limited for certain auxiliary sets. For example, DTD-Synthetic is a texture-style dataset which nominally contains 12 categories, but their backgrounds and appearances are highly similar; the lack of semantic diversity restricts the coverage of the image-level memory and weakens image-level classification.
>
> In contrast, when the auxiliary dataset contains only a single industrial category (e.g., SDD, KSDD2), the model has almost no opportunity to practice genuine cross-class retrieval during training. The anomaly prototypes then capture only one specific defect morphology and cannot form a transferable anomaly pattern, resulting in a noticeable performance drop for MRAD-FT. In more extreme cases when the auxiliary dataset is both single-category and strongly out-of-domain (e.g., medical datasets such as Kvasir and ClinicDB), the prototype distribution diverges significantly from industrial imagery, providing neither cross-class contrast nor anomaly patterns aligned with the target domain. Cross-domain retrieval thus becomes most difficult, and the degradation is the most severe.
>
> Overall, the experiments clearly show that MRAD does not rely on any specific auxiliary dataset. As long as the auxiliary source provides reasonable category and appearance diversity (such as VisA or BTAD), MRAD-FT and MRAD-CLIP maintain relatively stable performance under cross-dataset evaluation. Performance degradation mainly arises when the auxiliary dataset has extremely limited semantic coverage or exhibits large domain shifts, but even in these extreme settings the model does not collapse.

---

### Official Review · Reviewer_TyKg · 2025-11-11

**Soundness:** 3
**Presentation:** 2
**Contribution:** 2
**Rating:** 6
**Confidence:** 3

**Summary:**

This paper proposes MRAD, a memory-retrieval anomaly detection framework for zero-shot anomaly detection (ZSAD). Instead of learning a parametric mapping from features to labels, MRAD retrieves directly from a feature–label memory bank, avoiding training overhead and potential information loss. The authors design three variants: MRAD-TF, a training-free model using frozen CLIP encoders; MRAD-FT, which introduces two lightweight linear layers for fine-tuning retrieval metrics; and MRAD-CLIP, which incorporates region priors from MRAD-FT into CLIP’s learnable prompts for improved cross-modal alignment. Experiments on 16 industrial and medical datasets show that MRAD consistently outperforms state-of-the-art baselines such as AnomalyCLIP, FAPrompt, AdaCLIP, and WinCLIP, achieving superior performance with high efficiency and robustness.

**Strengths:**

1. MRAD reframes zero-shot anomaly detection as a non-parametric retrieval problem rather than a traditional model-fitting task. The proposed two-level (image- and pixel-level) memory bank is conceptually simple yet effective, marking a meaningful departure from existing CLIP-based prompt-learning approaches.

2. The experimental evaluation is extensive, covering 16 datasets across both industrial and medical domains. The results demonstrate the robustness and generalization ability of the proposed method.

**Weaknesses:**

1. While the empirical results are strong, the paper lacks theoretical justification or analytical insight into why a retrieval-based framework can outperform traditional parametric fitting approaches. The memory mechanism has been explored extensively in few-shot anomaly detection, and the novelty here lies in extending it to the zero-shot setting. Therefore, the authors should provide a more detailed discussion on why and how features extracted from the source domain can generalize effectively to target-domain detection tasks.

2. The overall reading flow of the paper could be improved, as certain sections are difficult to follow, and the mathematical formulations appear unnecessarily dense, which affects readability. The authors are encouraged to simplify equations where possible

3. It would be valuable to investigate whether AnomalyCLIP, when equipped with a similar “vanilla” memory mechanism, could achieve comparable performance. This comparison would help to more clearly demonstrate the effectiveness of the proposed memory-retrieval design, which is positioned as the main innovation of this work.

**Questions:**

See Weaknesses

---

> ### Author Response · Authors · 2025-11-19
> **Response to Reviewer TyKg (Part 1)**
>
> Thank you very much for your review and the insightful comments.
>
> **Q1: Analytical insight into why a retrieval-based framework can outperform traditional parametric fitting approaches and why/how source-domain features can generalize effectively to target-domain anomaly detection.**
>
> A1: Thank you for highlighting this important point. Traditional ZSAD approaches typically rely on a single auxiliary dataset such as VisA, training a classification or discriminative head to separate categories or templates as clearly as possible within the source domain. This design faces three inherent risks:
> (1) Overfitting: the classifier may over-rely on source-specific textures or background co-occurrences, which fail when the class or domain shifts;
> (2) Information compression: mapping a rich set of normal/anomalous patterns into a small set of classifier weights inevitably loses fine-grained structure, limiting coverage of diverse target-domain defects and impairing generalization.
>
> In contrast, we reformulate ZSAD as feature–label memory retrieval. Instead of compressing the VisA training set into a classifier head, we preserve all features and labels as explicit, queryable memory prototypes. Each stored prototype contributes a similarity score to the final decision, avoiding information collapse and ensuring that the entire distribution of normal/anomalous patterns remains accessible at test time. During training, each sample is required to retrieve against prototypes from all other classes rather than only from its own class. In other words, the query is always evaluated against a cross-class memory bank. This training setup explicitly simulates the “cross-category, cross-scenario retrieval” required during deployment and forces the model to extract a class-agnostic anomaly pattern, instead of relying on class-specific textures or contextual cues. When deployed on a new dataset, previously unseen categories simply act as new queries retrieving from the same memory bank. Since the retrieval mechanism at test time is identical to that during training, the matched prototypes remain aligned with this class-agnostic anomaly pattern, leading to robust cross-class and cross-domain generalization. We have added this explanation in Appendix A.2.
>
> To further illustrate why cross-dataset generalization is possible even in the train-free setting, Appendix C.2 provides a t-SNE visualization of prototypes from VisA (source) and MVTec (target). For MRAD-TF (Fig. 10(a)), we observe that in the embedding space induced by the pretrained CLIP vision encoder, normal and anomalous prototypes from different categories and datasets naturally separate along a certain direction within local neighborhoods, even though the global structure still exhibits strong class-wise clustering and is far from linearly separable. This indicates that CLIP has already encoded a weak but coherent anomaly pattern across categories: anomalous prototypes systematically shift away from normal ones in high-dimensional space. MRAD-TF leverages this property directly by storing all normal/anomalous prototypes and using cross-class similarity retrieval to judge whether a query deviates along this anomaly pattern. Because all prototypes contribute to the final similarity aggregation, this shared anomaly pattern learned by CLIP can be reused by target-domain anomalies, enabling MRAD-TF to achieve reasonable ZSAD performance on MVTec without any fine-tuning.
>
> In MRAD-FT (Fig. 10(b)), fine-tuning strengthens and disentangles this implicit anomaly pattern. During training, each sample is explicitly forced to retrieve across all other class prototypes, continuously simulating “category/domain shift” within the source domain. At the same time, a lightweight metric calibration (linear Q/K projection) aligns queries and memory into a shared subspace. The t-SNE visualization shows that, in this subspace, anomalous prototypes from different datasets cluster much more tightly into a single, class-agnostic group, while normal prototypes are pushed further away and retain only weak class structure. This demonstrates that fine-tuning has amplified the originally subtle anomaly signal into an explicit class-independent universal anomaly pattern. Consequently, when new target-domain categories appear, global retrieval against the same source memory remains consistent: the relative distances of query-to-prototype naturally align with this universal anomaly pattern, yielding more stable cross-class and cross-domain detection. The similarity analysis in Fig. 2 (Sec. 3.1.1) also confirms the increased separability before and after fine-tuning.

---

> ### Author Response · Authors · 2025-11-19
> **Response to Reviewer TyKg (Part 2)**
>
> **Q2: Concern about overall readability and dense mathematical formulations; some sections are hard to follow and encouraged to simplify equations where possible.**
>
> A2: Thank you for your suggestions regarding the writing and the readability of the formulas. In response to the comment about the mathematical expressions being too dense, we have streamlined the part related to memory-bank construction (Sec. 3.2.1), removing several non-essential equations and keeping only the core formulas that are necessary to understand the method. As for the issue of weak connections between sections, we fully agree with this feedback and will further refine the transitional paragraphs between related sections in the revised version, adding clearer guiding text in the method and experimental parts to improve the overall coherence of the narrative and the reading experience.
>
> **Q3: Whether AnomalyCLIP, when equipped with a similar “vanilla” memory mechanism, could achieve comparable performance, thereby more directly validating the effectiveness of the proposed memory-retrieval design.**
>
> A3: Thank you for this insightful suggestion, and we appreciate the attention to our memory-retrieval design as a key contribution of this work. In fact, MRAD-CLIP can be viewed as an enhanced variant of AnomalyCLIP with an added memory mechanism: while AnomalyCLIP relies on static learnable prompts, our approach injects the retrieval prior obtained from MRAD-FT into the prompt as a dynamic bias that adapts to each input image. This effectively equips AnomalyCLIP with a “vanilla” memory module without modifying its backbone architecture. As shown in the main experiments, this modification brings consistent improvements at both the image level and pixel level (average ROC gains of about 1.7% and 2.6%), demonstrating that memory retrieval indeed strengthens the discriminative capability of CLIP-based ZSAD frameworks. We agree with you that more systematic comparisons would be valuable, and in future work we plan to further explore finer-grained integration and ablation with AnomalyCLIP, as well as packaging the memory-retrieval design into a plug-and-play module for broader incorporation into existing zero-shot anomaly detection methods.

---

> > ### Comment · Reviewer_TyKg · 2025-11-24
> >
> > Thank the author for their effort in addressing my concerns. However, I am still worried about the novelty of the proposed memory mechanism. As mentioned above, the method combines AnomalyCLIP with the proposed memory mechanism, and the core contribution lies in this memory component. Therefore, a comparison between AnomalyCLIP with a vanilla memory bank and the proposed method is essential to demonstrate the effectiveness of the memory mechanism. This is an important ablation study. Otherwise, the true contribution of the paper remains ambiguous.

---

> > > ### Author Response · Authors · 2025-11-24
> > >
> > > Thank you for the additional clarification. We would like to first note that the architecture of AnomalyCLIP (including methods such as AdaCLIP and FAPrompt) is inherently difficult to integrate with a memory-bank mechanism. These AnomalyCLIP-like methods learn two prompt embeddings and compute anomaly scores based on the cosine similarity between these prompt embeddings and the patch embeddings of the test image. Its decision process is therefore driven by prompt–patch alignment.
> > > In contrast, a memory-bank approach relies on computing similarity among patch embeddings themselves, where the query patch is compared against stored patch embeddings from other images or categories. The two paradigms operate on different types of relationships: AnomalyCLIP evaluates how well a patch matches textual prompts, whereas a memory bank evaluates how a patch matches visual prototypes across instances. Due to this fundamental mismatch between prompt–patch alignment and patch–patch retrieval, AnomalyCLIP cannot be straightforwardly integrated with a memory-bank mechanism. Our contribution is precisely to introduce a retrieval-based memory-bank paradigm into the ZSAD setting, which is fundamentally different from the greedy nearest-neighbor strategies commonly used in FSAD and does not rely on instance-level correspondences.
> > >
> > > Regarding your concern on isolating the novelty of the memory-retrieval mechanism, our MRAD-FT variant directly provides such evidence. MRAD-FT shares with AnomalyCLIP only the CLIP visual feature extractor, and every part of its anomaly scoring relies exclusively on non-parametric similarity retrieval over the memory bank. It does not use prompt learning, prompt-based scoring, or any architectural component of AnomalyCLIP beyond the backbone. In other words, MRAD-FT represents a pure retrieval-based instantiation and demonstrates the discriminative power of the memory paradigm itself. Across most datasets (Sec. 4.2, Tab. 1), MRAD-FT consistently exceeds AnomalyCLIP by a notable margin, which confirms that memory-driven retrieval is an effective and independent mechanism for ZSAD.
> > >
> > > We wonder whether our explanation already addresses your concern regarding the novelty of the proposed memory-retrieval paradigm. If not, we would greatly appreciate it if you could clarify what form of “AnomalyCLIP with a vanilla memory bank” you have in mind, and we would be happy to include additional experiments accordingly.

---

> ### Comment · Reviewer_TyKg · 2025-11-26
>
> The simplest implementation is similar to WinCLIP, which defines a patch score to measure the discrepancy between the query image and reference images. The reference images can be selected from the training images, as done in this paper. The final zero-shot result is then obtained by integrating the zero-shot score with the patch score

---

> > ### Author Response · Authors · 2025-11-27
> >
> > Thank you very much for the additional clarification. Following your suggestion, we implemented a simple “AnomalyCLIP + vanilla memory bank’’ baseline. Our implementation follows the idea of WinCLIP’s few-shot reference module, but adapted to the zero-shot setting. We keep AnomalyCLIP’s original zero-shot score (prompt–patch alignment). In addition, we reuse the pixel-level memory bank as the reference patch set (built on VisA). For each query patch, we compute a nearest-neighbor discrepancy against this reference set and integrate this discrepancy score with AnomalyCLIP’s zero-shot prediction. This matches the simplest and most direct form of the baseline you requested.
> >
> > VanillaMB denotes “Vanilla Memory Bank”. The pixel-level P-AUROC results are:
> > | Dataset  | AnomalyCLIP | AnomalyCLIP + VanillaMB | MRAD-FT | MRAD-CLIP |
> > | -------- | ----------- | ----------------------- | ------- | --------- |
> > | MVTec    | 91.1        | 91.0                    | 92.2    | 93.0      |
> > | BTAD     | 93.3        | 89.2                    | 94.7    | 95.4      |
> > | MPDD     | 96.2        | 96.5                    | 97.4    | 97.9      |
> > | SDD      | 90.1        | 90.3                    | 91.0    | 93.0      |
> > | KSDD2    | 97.9        | 98.2                    | 98.8    | 98.9      |
> > | Kvasir   | 81.9        | 82.0                    | 83.9    | 84.3      |
> > | ClinicDB | 85.9        | 85.3                    | 85.9    | 87.3  |
> >
> >
> > We find that adding this vanilla memory bank to AnomalyCLIP does not bring meaningful improvement. On some datasets (e.g., BTAD, ClinicDB), the performance even drops. We believe this mainly happens because nearest-neighbor patch retrieval works in FSAD, where reference and query images usually share the same object or background, but it does not transfer to ZSAD, where categories differ and cross-dataset patch matching becomes unreliable.
> >
> > In contrast, both MRAD-FT and MRAD-CLIP outperform this baseline on all datasets. This is because MRAD stores explicit feature–label pairs and supports cross-category, cross-dataset retrieval, which is important for generalizable zero-shot anomaly detection. These results further support the effectiveness of the proposed memory-retrieval approach.

---

### Author Response · Authors · 2025-11-21
**General Response and Revision Summary**

Dear Area Chair and Reviewers,

We sincerely thank you for your time and constructive comments. Below we summarize the main revisions made in response to the reviewers:

- For Reviewer **TyKg**’s concern regarding the core mechanism of MRAD, we added detailed explanations and visualizations, and implemented the suggested new baseline "AnomalyCLIP + vanilla memory bank" to further demonstrate the novelty of our memory-retrieval design.

- For Reviewer **iQze**’s questions on hyperparameters and the memory-bank module, we added the corresponding ablation studies covering projection depth, two-level memory, and sensitivity to ρ / Top-K / thresholding.

- For the concern jointly raised by Reviewers **iQze** and **toN8** about the choice of auxiliary datasets, we conducted extensive auxiliary-dataset replacement experiments and analyzed how memory bank quality and diversity affect cross-domain retrieval.


- For the concerns jointly raised by  Reviewers **toN8** and **LY8o** regarding prompt design, fine-tuning efficiency, and memory size, we provided additional quantitative comparisons and clarifications, and further discussed memory quality, diversity requirements, and known limitations of MRAD.

- For Reviewer **LY8o**’s concern about robustness, we added experiments on random seeds and distribution shifts to verify the stability of MRAD. In addition, to address the reviewer’s question regarding supervision requirements, we further clarified the minimal supervision needed to build the memory bank and included experiments using synthetic anomalies.



We have revised the paper to address all concerns and have provided detailed, point-by-point responses to each reviewer’s comments. The main changes are summarized as follows:

- **Sec. 3.2.1:** Simplified the description of memory-bank construction and removed redundant equations.
- **App. A.2, C.2:** Added a concise mechanism explanation and t-SNE visualizations to illustrate cross-domain retrieval.
- **App. C.3:** Added parameter/memory/training-time/latency comparisons to quantify the “lightweight” claim.
- **App. C.4:** Added an ablation on the two-level memory bank by removing image-level or pixel-level branches.
- **App. C.5:** Added auxiliary-dataset replacement experiments to analyze robustness to the choice of source data.
- **App. C.6:** Expanded medical ZSAD evaluation with an additional medical dataset.
- **App. C.7:** Added stability and robustness analysis of our method.
- **App. C.8:** Added experiments with “normal data + synthetic anomaly patches” to probe supervision requirements.
- **App. C.10:** Added an additional baseline (“AnomalyCLIP + vanilla memory bank’’) to further isolate and validate the contribution of the proposed memory-retrieval design.

We appreciate the reviewers’ thoughtful comments and hope these revisions improve the clarity and strength of the paper. Detailed responses are given below.

---

### Meta-Review · Area_Chair_YGtU · 2026-01-06

**Summary:**

Reviewers’ concerns included theoretical justification for the retrieval-based framework, validation of MRAD’s novelty, and missing ablations. They also questioned auxiliary dataset, insufficient medical dataset coverage, hyperparameter sensitivity, and the quantified validity of the "lightweight" claim. Other concerns come from the scalability

Additionally, concerns arose about scalability,  minimal supervision requirements for memory construction, etc. The authors have addressed these by adding theoretical explanations, targeted experiments, and quantitative analyses, confirming the method’s soundness and contribution.

**Reviewer Concerns:**

Most of the concerns seems have been addressed, for example:
1. Theoretical justification for retrieval-based superiority
2. Readability & dense formulas
3. AnomalyCLIP + vanilla memory bank comparison
4. ablation on the two-level memory bank
5. Alternative auxiliary datasets

No outstanding concerns remains to me.

**Reviewer Scores:**

Reviewer TyKg: the score remains unchanged
Reviewer iQze: can be increased to 7 as the authors have provided extra experiments.
Reviewer toN8: extra datasets and comparisions are provided, may increase to 7
Reviewer LY8o: the concerns of memory and scalability were discussed and random seeds experiment was provided, as well as the Multi-scale priors tests. may increase to 7

---

### Decision · Program_Chairs · 2026-01-26

Accept (Poster)